# A systematic review and meta-analysis of the relationship between subjective interoception and alexithymia: Implications for construct definitions and measurement

**Kristen Van Bael** *, **Jessica Scarfo**, **Emra Suleyman**, **Jessica Katherveloo**‡, **Natasha Grimble**‡, **Michelle Ball**

Institute for Health and Sport, Victoria University, Melbourne, VIC, Australia

☯ These authors contributed equally to this work.
‡ JK and NG also contributed equally to this work.
* kristen.vanbael@vu.edu.au

## Abstract

Although research indicates that self-reported interoception is associated with deficits in identifying and describing emotional experience, and externally oriented thinking styles (alexithymia), this relationship appears moderated by how interoception is measured. A systematic review and meta-analyses examined the association between self-reported interoception and alexithymia, investigating how different interoceptive questionnaires relate to alexithymia at global and facet levels. PsychINFO, PubMed, Scopus, and Web of Science databases were searched with predefined terms related to self-reported interoception and alexithymia. Three reviewers independently assessed articles, extracted data, and undertook risk of bias assessment. Thirty-two cross-sectional studies published between 1996 and 2023 were included. Random-effects meta-analyses and narrative synthesis indicated that global alexithymia was positively associated with measures of interoceptive confusion, autonomic nervous system reactivity, and heightened interoceptive attention, and inversely associated with interoceptive accuracy and adaptive interoception, indexed by composite Multidimensional Assessment of Interoceptive Awareness scores, but particularly interoceptive trusting, self-regulation, and attention regulation. These patterns were observed for alexithymic facets and stronger in magnitude for difficulty identifying feelings and difficulty describing feelings, relative to externally oriented thinking. Overall, results suggested that the association between self-reported interoception and alexithymia differs as a function of the interoceptive self-report. The review highlighted issues with construct definition and operationalisation and determined that existing interoceptive self-reports broadly capture maladaptive and adaptive sensing, attention, interpretation, and memory. The findings underscore the importance of specifying interoceptive constructs and using appropriate assessments to improve convergence between constructs and measurements, further suggesting potential clinical utility in using existing self-reports to measure interoception and alexithymia, facilitating interventions targeting mind-body connections.

**Data Availability Statement:** We confirm that the submission contains all raw data required to

replicate the results of the meta-analyses and systematic review. They are available in Files S4 and S7 to S9 of the Supporting Material submitted with the revised manuscript. The data are freely accessible, as all extracted data for the study are provided in published studies and/or pre-print articles.

**Funding:** The author(s) received no specific funding for this work.

**Competing interests:** The authors declare that no competing interests exist.

## Introduction

Alexithymia represents a multifaceted trait typified by diminished capacities for identifying and describing emotions, which may be accompanied by tendencies for focusing on features of the external environment [1–3]. Such deficits are associated with various maladaptive processes and outcomes, including somatisation [4], emotion regulation strategies characterised by avoidance and withdrawal [5], inefficient coping [6], and heightened physiological and psychological stress [7]. Whilst traditional views of alexithymia propose the involvement of cognitive and emotional deficits [2], contemporary proposals place deficits in the perception and integration of internal bodily signals—interoception—at its core [8]. Such views postulate that impaired awareness and interpretation of ongoing sensations from within the body may coincide with diminished recognition, articulation, and experience of emotions.

Interoception encompasses unconscious and conscious experiences of internal bodily signals, crucial for maintaining homeostasis and wellbeing [9, 10]. The psychological context in which interoceptive stimuli are processed influences ongoing perceptions and adaptive responsivity [11], wherein efficient physiological regulation necessitates conscious attention and accurate interpretation of the signal [12, 13]. These signals are transmitted from peripheral systems to the insula [10], supporting emotional, mental, and physical wellbeing—core components of the mind-body connection [12, 14]. Adaptive interoceptive processing accordingly entails healthy attention to signals and accurate detection of their meaning in context, which can facilitate behaviours aimed at maintaining physiological integrity, particularly during states of felt perturbation [13, 15, 16]. This is contrasted with maladaptive processing, entailing dysfunctional attention (e.g., hypervigilance, avoidance), and impaired accuracy, which may hinder effective regulation and adaptive decision-making. There is recognition that strongly held interoceptive beliefs can influence clinical symptoms [17]; moreover, that self-reports capturing interoceptive beliefs and interpretations may yield greater insight into clinical status than brain-based or behavioural measures [17, 18]. Indeed, various conditions have increasingly been characterised by atypical interoceptive and emotional processing, including anxiety [19], autism spectrum disorder [12, 20], feeding and eating disorders [21, 22], depression [13, 19], and somatic symptom and related disorders [23, 24]. As such, examining the relationship between subjective interoception, measured by self-report, and alexithymia may facilitate the delivery of targeted interventions and treatments aimed at cultivating adaptive mind-body integration.

Trevisan et al. [25] previously meta-analysed the association between self-reported interoceptive constructs and global alexithymia. Although significant associations were identified, previous findings are ultimately clouded by both a lack of convergence between interoceptive constructs and employed measures in research and the consideration of alexithymia as a global characteristic. Whilst efforts to elucidate key differences between interoceptive self-report scales have commenced [e.g., 12, 26, 27], consideration of these differences in the context of alexithymia at global and facet levels has yet to be operationalised. This study therefore employs a systematic review and meta-analytic approach to examine the association between specific interoceptive self-report scales and alexithymia at global and facet levels.

Interoception is now conceptualised as a multidimensional function. Prior to 2015, various terminologies and measurements for interoceptive processes existed. To encourage consistency in construct definitions and operationalisation of interoceptive abilities, Garfinkel and colleagues [28] proposed a parsimonious three-dimensional model of interoception. 'Interoceptive accuracy' was defined as involving objective accuracy in detecting internal bodily sensations, with performance gauged via behavioural paradigms, such as heartbeat detection or discrimination tasks. By contrast, the purely subjective 'interoceptive sensibility' construct was

defined as the "self-perceived dispositional tendency to be internally self-focused and interoceptively cognisant" (p. 67), where self-reports probing perceived aptitude, such as the Body Perception Questionnaire (BPQ) [29], were recommended for assessment. Lastly, 'interoceptive awareness' was defined as metacognitive awareness of accurate detection of internal bodily sensations, where measurement involved the correspondence between objective performance and subjective performance appraisal.

Alternative interoceptive taxonomies have since been proposed, similarly demarcating objective and subjective interoceptive dimensions. With respect to subjective dimensions, various constructs have been suggested, including 'interoceptive sensibility' [9, 28, 30], 'interoceptive self-report scales' [9], 'self-reported interoceptive attention', 'self-reported interoceptive accuracy' [27], and 'self-report of interoception and beliefs' [17]. To date, Garfinkel and colleagues' [28] three-dimensional framework is most frequently cited across the literature [26, 31] despite evidence indicating that this model should be revised [27, 31]. Moreover, although 'interoceptive sensibility' was initially specified to involve self-perceived tendencies for focusing on and detecting bodily sensations, consistency in how the construct is operationalised via self-report is notably lacking in research [26, 32].

Interoceptive self-report scales are proposed to generally assess: i) self-reported dispositional tendency of attention toward bodily signals relevant to homeostatic needs, such as hunger, fatigue, illness, and injury, and to some degree, emotional arousal, and ii) self-perceptions of accuracy in the discrimination and interpretation of such signals [12]. A systematic review identified that frequently administered measures, including the multifactorial BPQ [29, 33], the eight-scale Multidimensional Assessment of Interoceptive Awareness (MAIA) [34, 35], Body Awareness Questionnaire (BAQ) [36], and Self-Awareness Questionnaire (SAQ) [37], tend to be subsumed under the 'interoceptive sensibility' umbrella term in the literature [26]. Recent findings reveal major issues with this approach, as the measures do not converge and assess relatively distinct constructs [26, 32, 38]. Furthermore, evidence indicates that the MAIA measures may be better conceptualised as capturing three interoceptive constructs (adaptive interoception, interoceptive not-distracting, interoceptive not-worrying) rather than eight [26, 32, 38, 39]. Considering these factors, operationalisation appears to be in a state of relative detachment from interoceptive construct definitions [26, 31]. Such discrepancies pose a major challenge to empirical interpretation and complicate future replication attempts.

These issues are exemplified by previous meta-analytic findings reported by Trevisan et al. [25] concerning the association between interoceptive dimensions and global alexithymia. They determined that alexithymia was inversely associated with 'subjective interoceptive accuracy'; however, analysis for this construct consisted of pooled self-reports capturing neutral interoceptive accuracy and inaccuracy—some which negatively correlate [40]. Meta-analysis further demonstrated no overall significant relationship between 'interoceptive sensibility' and alexithymia when interoceptive self-report data were aggregated. Additional analysis, however, indicated that this relationship is significantly moderated by employed measures. Specifically, the BPQ and alexithymia were positively associated, such that heightened awareness of internal bodily sensations related to higher alexithymia. By contrast, averaged Noticing and Emotional Awareness MAIA subscales and alexithymia were negatively associated. In other words, greater awareness of comfortable, uncomfortable, and neutral body sensations (Noticing scale) and the connection between bodily sensations and emotion (Emotional Awareness scale) were related to lower alexithymia [25]. As such, subjective interoception is seemingly related to alexithymia, but the direction and strength of this association depends upon the construct measured by the administered interoceptive self-report measure.

Extending on these findings within the context of differentiated interoceptive styles [11], Trevisan et al. [12] proposed that the MAIA captures healthy, adaptive interoceptive attention

styles that facilitate regulatory behaviour, and adequately differentiates between adaptive and maladaptive attention. Conversely, the BPQ was proposed as capturing maladaptive attention, characterised by anxiety-driven hypervigilance toward bodily sensations and somatisation. Furthermore, that the BAQ potentially taps into maladaptive interoceptive attention, given that items assess sensitivity to body cycles and rhythms, detection of subtle deviation in typical functioning, and anticipation of body reactions in addition to self-reported attentiveness to normal non-emotional bodily processes. Relatedly, Murphy et al. [27, 40] have offered specific measures, according to whether subjective accuracy or attention is the interoceptive construct of interest. They recommended employment of the BPQ to measure interoceptive attention beliefs and the Interoceptive Accuracy Scale (IAS) or Interoceptive Confusion Questionnaire (ICQ) to assess interoceptive accuracy beliefs. More recently, Desmedt et al. [31] proposed a hierarchical framework consisting of interoceptive factors, which are comprised of subfactors denoting what aspect is being measured, according to specific interoceptive self-report scales. This includes 'interoceptive sensing' (the sense of internal signals by the nervous system across conscious and nonconscious levels; e,g, IAS), 'interoceptive attention' (any attentional process related to internal signals; e.g., MAIA-Attention Regulation, MAIA-Not-Distracting, Interoceptive Attention Scale), 'interoceptive interpretation' (any interpretation, belief, attitude, and categorisation of internal signals; e.g., MAIA-Emotional Awareness, MAIA-Not-Worrying, MAIA-Trusting, Somatosensory Amplification Questionnaire), and 'interoceptive memory' (any memory process related to internal signals).

Considered together, there are key differences in existing interoceptive self-report measures proposed to capture overarching constructs, including subjective interoceptive accuracy, attention, and interpretation. However, various self-report scales used in interoceptive research have not been thoroughly considered. Determining what these key differences are, based on their relationships with alexithymia and in the context of extant frameworks, may facilitate identification of a construct validity framework and promote employment of measures that capture specific interoceptive constructs, thereby enhancing the validity and reliability of clinically meaningful interpretations.

Interest in subjective interoception is increasing, as evidenced by the development of self-report scales measuring interoceptive perceptions and beliefs, including the ICQ [41], IAS [40], Interoceptive Attention Scale (IATS) [42], Interoceptive Sensory Questionnaire (ISQ) [43], Interoceptive Sensitivity and Attention Questionnaire (ISAQ) [44], and Three-Domain Interoceptive Sensations Questionnaire (THISQ) [45]. Although previous meta-analytic findings provide evidence for some adaptive and maladaptive interoceptive percepts linked to global alexithymia, analysis of interoceptive scale subsets has been limited to the BPQ and two MAIA scales [25]. As such, the relationship between subjective interoception, as measured by various self-report scales, and alexithymic facets beyond global measurement requires further interrogation. Alexithymia involves various constituents that differentially affect treatment outcomes [46]. If particular interoceptive constructs and beliefs—captured via self-report—indeed relate to particular alexithymic facets, then clarifying these relationships may shed light on interoceptive and alexithymic mechanisms that could be targeted in mind-body therapies.

We heed warnings regarding issues with aggregating these data for the purposes of meta-analysis [26, 31]. However, it is of theoretical interest to examine how specific interoceptive self-report measures relate to alexithymia. The aim of this pre-registered systematic review and meta-analysis was to examine the relationship between various interoceptive self-report scales and alexithymia. This was adopted to clarify key differences in interoceptive self-report scales contributing to different relationships with alexithymia [12]. Through this, we anticipated identifying which interoceptive measures could broadly be conceptualised as tapping into 'adaptive' and 'maladaptive' interoceptive self-appraisals, based upon how the scales relate to

alexithymia. This could assist with improving construct definitions and operationalisation of self-reported interoception, as meta-analytic approaches should enable the quantification of associated constructs in conjunction with recommendations for suitable measurements. Further, examining an empirical association between subjective interoception and alexithymia may bolster arguments for considering interoception in the measurement of alexithymia, particularly in the context of co-occurring clinical conditions [47]. Accordingly, the following research question was developed:

*Does the relationship between subjective interoception and alexithymia differ as a function of different interoceptive self-report scales?*

## Methods

### Design

The systematic literature search was conducted according to the 2020 Preferred Reporting Items for Systematic Review and Meta-Analyses (PRISMA) guidelines [48; see S1 File]. The protocol was registered on the PROSPERO database (identification number: CRD42023437654), accessible online at https://www.crd.york.ac.uk/prospero/display_record. php?ID=CRD42023437654.

Following data extraction for the systematic review, we identified that meta-analysis was more appropriate. As the PROSPERO record could not be updated to reflect this change in analytic approach, the meta-analysis protocol was pre-registered at https://osf.io/3nsyc/.

### Search strategies

A search strategy was developed using the following terms: (interoceptive sensibility OR interoceptive self-report OR self-reported interoception OR subjective interoception OR interoceptive evaluation OR interoceptive beliefs OR subjective interoceptive attention OR subjective interoceptive accuracy OR self-reported interoceptive attention OR self-reported interoceptive accuracy) AND (alexithymia or alexithymic or alexithym*). The search was conducted from 13 July to 01 October 2023, and restricted wherever possible to titles, abstracts, and key words identified within PsychINFO, PubMed, Scopus, and Web of Science databases. Additional sources were identified through reference list screening for included articles and Google Scholar. All sources were imported into Covidence, a web-based collaboration software platform that streamlines the production of systematic and other literature reviews.

### Eligibility criteria

**Inclusion criteria.** Studies were included if they: (i) were written in English; (ii) had samples aged 18 years and older; (iii) were published in a peer-reviewed journal or uploaded to a pre-print database that was accessible through online databases through the authors' institutional library or via interlibrary loans; (iv) measured subjective interoception using a validated self-report scale; (v) measured alexithymia using a validated self-report scale; (vi) quantitatively measured the relationship between subjective interoception and alexithymia; and (vii) reported statistics that enabled interpretation of the strength and direction of the relationship between subjective interoception and alexithymia. Studies examining clinical samples were considered, as the previous meta-analysis identified a moderating effect for clinical conditions [25].

**Exclusion criteria.** Studies were excluded if: (i) the sample was aged 17 years or younger; (ii) the interoceptive self-report scale was deemed to assess constructs other than subjective

interoception; (iii) measurement of alexithymia was not a validated self-report scale; and (iv) the study did not report statistics that enabled interpretation of the relationship between subjective interoception and alexithymia.

**Data extraction.** A data extraction form was developed in Covidence. Three reviewers (KVB, JK, NG) independently extracted the following data: author and year; country; study design; sample characteristics; sample size; percent of sample identifying as female; clinical status; investigated interoceptive construct; interoceptive self-report scale; analysed interoceptive scales; alexithymia self-report; analysed alexithymia scales; covariates or controlled variables; results; effect sizes (correlation coefficients–*r*). Extraction was completed in Covidence and exported to Excel. We did not contact authors for unreported effect sizes, based on described poor response rates for previous meta-analyses [49, 50]. Disagreements between reviewers were resolved through open discussions to reach consensus.

## Analysis approach

**Primary meta-analyses.** See https://osf.io/ky3qf for the pre-registered analysis plan. Correlation coefficients were extracted from each study to represent the effect size magnitude of the relationship between self-reported interoception and alexithymia. We performed Fisher's *Z* transformations in Excel using the following equation on *r* values to improve normality:

$$z = 0.5 \times \ln\left(\frac{1+r}{1-r}\right)$$

Variance of Fisher's *Z* was calculated in Excel as:

$$Vz = \frac{1}{(n-3)}$$

Fisher's *Z* was converted back to *r* values in text and tables for summary statistics, whereas Fisher's *Z* transformed values are reported in figures at https://osf.io/3nsyc/. Per Trevisan and colleagues' [25] approach, we did not convert standardised betas to simple correlations as this may introduce bias in summary effect size estimation.

Considering evidence demonstrating that measures of self-reported interoception do not assess the same construct [26, 39], we did not aggregate interoceptive data. Data were split according to interoceptive self-report scale. Separate meta-analyses were conducted according to specific interoceptive scales and four alexithymic outcomes (global alexithymia, difficulty identifying feelings, difficulty describing feelings, and externally oriented thinking) where the number of effects (*k*) per scale and alexithymic outcome was ≥2. In several instances, the same participants from a study contributed multiple effect sizes (i.e., studies included more than one alexithymia scale). In line with the method of Robinson et al. [50], we divided the total number of study participants by the number of effect sizes the study contributed to the meta-analysis to provided adjusted sample sizes. Several studies reporting findings from the same samples were also noted. Only one of the study's effects were included to ensure independence of observations.

The meta-analyses were conducted using SPSS Statistics, Version 29. The conservative Sidik-Jonkman estimator was applied. Random effects models were employed for all analyses, which assumes that variability in effect sizes among studies may vary between studies due to heterogeneity, including sample characteristics [51]. Heterogeneity was investigated for each meta-analysis, and *Q* and *I*$^2$ statistics are reported. Publication bias was also assessed. Funnel plots were produced to show the relationship between effect sizes and standard error. Egger's tests were used to assess the asymmetry of the funnel plot where there were sufficient studies

included ($k \geq 10$). Influence analysis using the leave-one-out method was used to assess the influence of individual studies on pooled effect sizes. Cohen's [52] recommendations were used for effect size interpretation (weak: $r = 0.00$ to $0.29$, moderate: $r = 0.30$ to $0.49$, strong: $r \geq 0.50$). Pooled effects were interpreted as significant where $p < .05$.

**Secondary analyses.** If outliers or influential effects were identified in primary analyses, they were removed for secondary analyses. Where high heterogeneity was observed, we conducted subgroup analysis according to clinical status, as this was expected to contribute to variance [25]. Some studies pooled clinical and typically developed participants into a single group for correlational analyses. Following the approach of Trevisan et al. [25], we considered these samples as representing a clinical category to maximise statistical power. Subgroup analysis was conducted where indicated, based on the region where the sample was located (Asia, Australasia, Europe–UK, Europe–Other, North America), as several interoceptive self-reports, such as the MAIA, have been translated from English into other languages [35] and culture can shape interoceptive and emotional conceptualisations [e.g., 53, 54]. As global alexithymia was indexed using three scales across the included studies, we conducted subgroup analysis according to alexithymic measure.

**Risk of bias assessment.** To assess for risk of bias, the Strengthening the Reporting of Observational studies in Epidemiology (STROBE) checklist for cross-sectional studies [55] was utilised. The three reviewers independently evaluated all studies for risk of bias based on whether the 22 STROBE checklist items indicated low, medium, or high risk of bias. Articles, online supplements, and study pre-registrations were consulted for these assessments. Disagreements were resolved through consensus. For reviewer agreement regarding risk of bias, see the S2 File. The S3 File contains assessment results with ratings according to STROBE items.

## Results

### Study selection

The search parameters yielded the following (Fig 1).

The search strategy identified an initial 232 articles via database searching. Following Covidence removal of duplicates ($n = 97$), the remaining 135 studies were screened against title and abstract. Following title and abstract screening, 103 papers were collectively deemed as not meeting inclusion criteria; 31 papers subsequently remained and were assessed for full-text eligibility. Fifteen studies were excluded and a total of 16 studies identified through database searches were included. Google Scholar searches yielded 797 results. Of these, 58 studies were retrieved. Eleven studies identified via Google Scholar were deemed eligible and relevant; included studies then totalled 27. Reference lists for these studies were scanned for additional sources, of which five were identified. These were screened against eligibility criteria and included. We identified that one study was ineligible following screening, as results concerned the same sample [56]. To maintain independence of observations, the study was excluded. The number of articles included in final reporting was therefore 32. At all review stages, fair consistency was observed amongst reviewers ($>65\%$), with conflicts resolved through discussions. Inter-rater reliability amongst reviewers is provided in the S2 File.

### Study characteristics

Full study characteristics are reported in the S4 File. Publication dates ranged from 1996 [57] to 2023 [58–63]. Most studies were conducted in Western countries, most frequently in the United Kingdom ($n = 10$). Most studies used a cross-sectional design ($n = 35$); one used

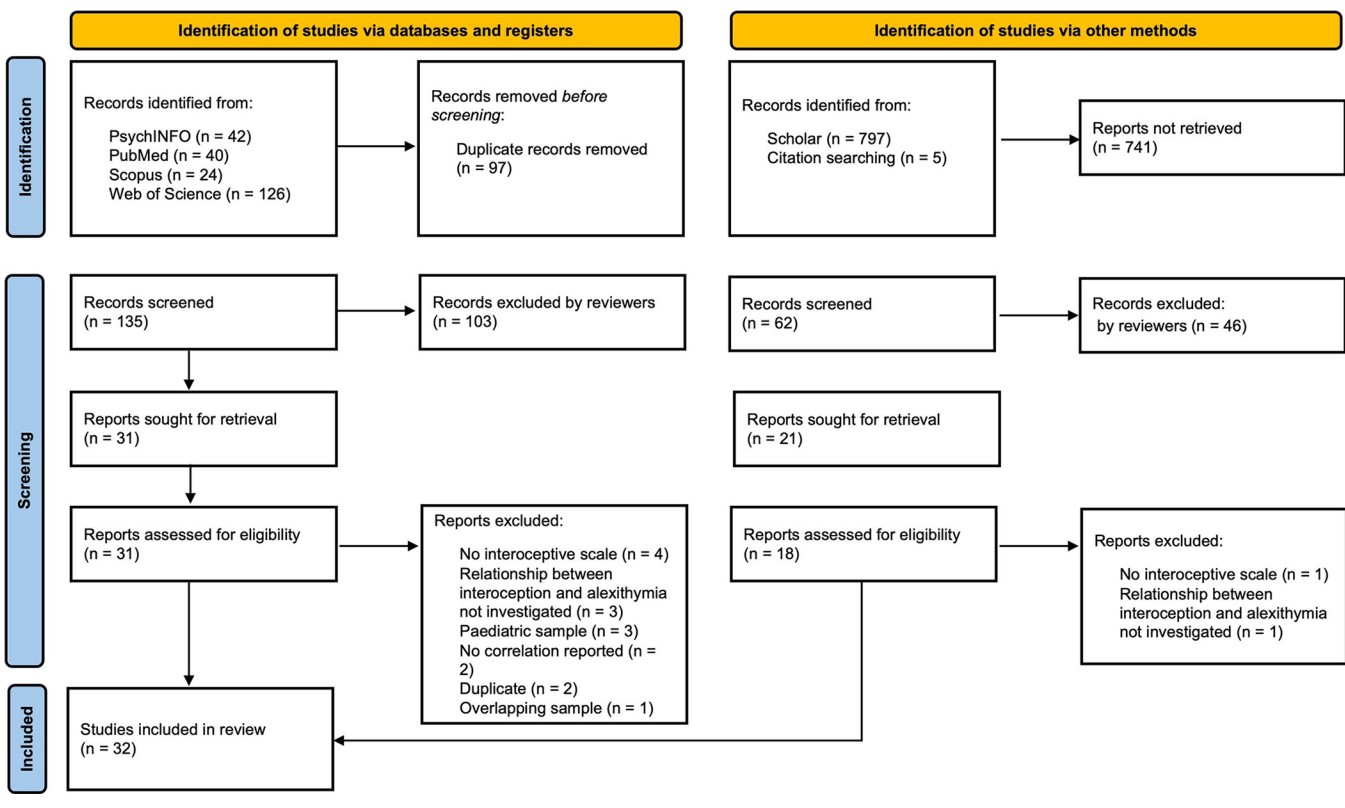

**Fig 1. PRISMA flow diagram of study inclusion.**

randomised experimental design and conducted a cross-sectional analysis of self-reported interoception and alexithymia following mood induction [64].

Overlapping samples were noted for three studies based in Germany and Austria [65–67]. All studies included students from the University of Potsdam recruited by Ventura-Bort et al. [67]; findings reported by Brand et al. [65] and Tünte et al. [66] pertained to the same Vienna and Potsdam samples, where employed materials were similar. To ensure independence of observations, the decision was made to include Brand et al. [65] findings for the MAIA, BPQ, ICQ, BPQ, IAS and alexithymia in the meta-analyses. Tünte et al. [66] however, reported findings for the IATS with alexithymia in two samples. As such, these IATS findings were included in analysis. Findings for a principal components analysis (PCA) of interoception and alexithymia scales reported by Ventura-Bort [67] was included in narrative synthesis.

## Participants

Across the 32 included studies, the total number of participants was 7819 with a minimum number of 18 [68] and a maximum number of 759 [63]. Most studies recruited males and females (91.67%), although women tended to be disproportionately represented. The remaining studies recruited women [64, 69] or men only [58]. Participants' ages ranged from 18 to 91 [40]. Most commonly, samples were non-clinical, and drawn from general communities or primarily university students and/or staff. With respect to clinical conditions of interest, autism spectrum disorder (ASD) was most frequently investigated ($n$ = 4) [43, 58, 59, 70]. The relationship between subjective interoception and alexithymia was also examined in patients with anorexia nervosa (AN) [57], fibromyalgia [71], functional motor disorders (FMD) [72],

irritable bowel diseases (IBD) such as Crohn's disease (CD) and ulcerative colitis (UC) [62], and individuals who self-reported an existing psychiatric diagnosis [40].

## Investigated interoceptive constructs

Various interoceptive constructs were investigated across the included studies. In order of frequency, these were: 'interoceptive sensibility' ($n = 12$), 'interoceptive awareness' ($n = 11$), 'self-reported interoceptive accuracy' ($n = 4$), 'self-reported interoceptive attention' ($n = 3$), 'interoception' ($n = 1$), 'interoceptive challenges' ($n = 1$), 'interoceptive confusion' ($n = 1$), 'interoceptive impact' ($n = 1$), 'self-reported interoception' ($n = 1$), 'and 'subjective interoception' ($n = 1$). Where umbrella terms such as 'sensibility' were employed, consistency in construct operationalisation was notably lacking. By contrast, investigation of more specific constructs, such as 'self-reported interoceptive accuracy', related to administration of specific measures.

Since at least 2018, there was an observed trend toward consistent application of the term 'interoceptive sensibility', as defined by Garfinkel et al. [28]. 'Interoceptive awareness' was also investigated frequently, despite the tendency for researchers to ascribe the 'sensibility label' to self-reported interoception. This prompted evaluation of reasons for using the term. Of the 11 studies investigating this construct, three predated publication of the 3-factor framework from Garfinkel et al. [28] that proposed 'interoceptive sensibility' [37, 57, 68]. Five studies that employed the MAIA defined 'interoceptive awareness' based on Mehling and colleagues' [34, 35] conceptualisations [63, 70, 73–75] in contrast to the 'interoceptive awareness' domain proposed by Garfinkel et al. [28]. One study cited Craig [10] in defining 'interoceptive awareness' as "the conscious perception of internal bodily states that emanate from the autonomic nervous system" [76]. One study cited Khalsa et al. [9] to define the construct as "the perception and integration of signals related to body states" [64]. One study drew on previous findings to define 'interoceptive awareness' as "the capacity to attune to physiological experiences" [61].

More recent studies investigated self-reported interoceptive accuracy and attention, as proposed by Murphy et al. [27], employing self-reports assessing these constructs [40, 65, 66, 77]. Several papers proposed alternative terms that capture specific interoceptive constructs according to what the self-report scale measures, such as 'interoceptive impact', proposed to encompass the "the influence of interoception on everyday life" [78] and 'interoceptive challenges', involving impaired processing of "interoceptive signals that report the moment-to-moment condition of the body" [43].

## Administered measures

**Interoceptive self-report scales.** Most studies administered one scale to measure self-reported interoception ($n = 26$); six studies administered multiple measures [40, 65–67, 79, 80]. An overview of the interoceptive self-report scales employed in included studies, including acronyms and scale descriptions, is provided in the S5 File.

The Multidimensional Assessment of Interoceptive Awareness (MAIA) and MAIA, Version 2 (MAIA-2) was most frequently employed to measure self-reported interoceptive constructs ($n = 18$). Other measures included the Body Perception Questionnaire (BPQ; $n = 7$), Interoceptive Accuracy Scale (IAS; $n = 6$), Interoceptive Confusion Questionnaire (ICQ; $n = 5$), Body Awareness Questionnaire (BAQ; $n = 2$), Interoceptive Sensory Questionnaire (ISQ; $n = 2$), the Interoceptive Awareness scale of the Eating Disorder Inventory (EDI-IAw $n = 1$), Interoceptive Attention Scale (IATS; $n = 1$), Self-Awareness Questionnaire (SAQ; $n = 1$), Sensory Profile: Interoception (SPI; $n = 1$), and Three-Domain Interoceptive Sensations Questionnaire (THISQ; $n = 1$). Although Brewer et al. [41] reported mixed psychometric evidence for the ICQ, these findings were included in the present review, as other studies

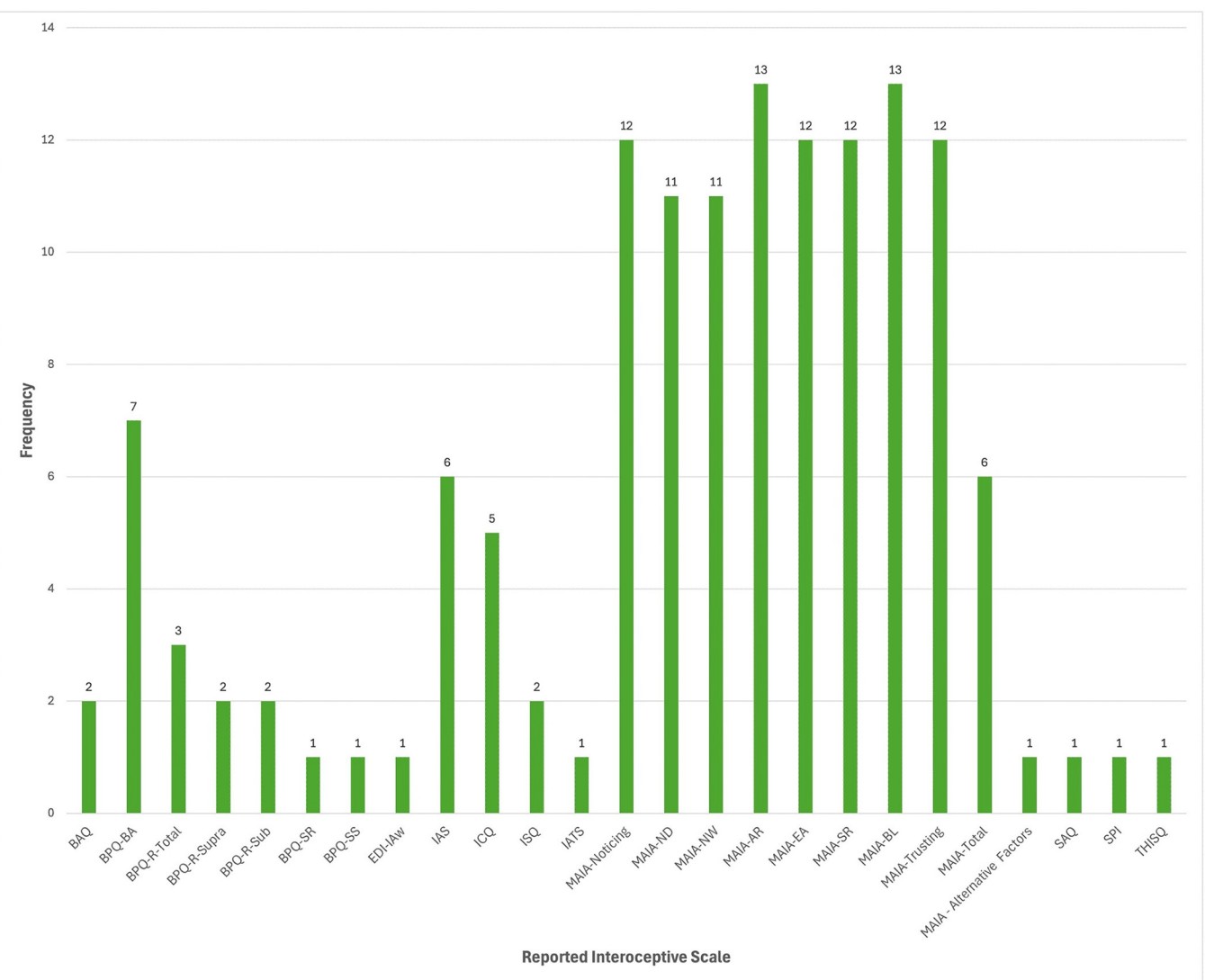

**BAQ**: Body Awareness Questionnaire; **BPQ**: Body Perception Questionnaire; **BPQ-BA**: Body Awareness Scale; **BPQ-R-Total**: Autonomic Reactivity Scale; **BPQ-R-Supra**: Autonomic Reactivity - Supradiaphragmatic; **BPQ-R-Sub**: Autonomic Reactivity - Subdiaphragmatic; **BPQ-SR**: Stress Response; **BPQ-SS**: Stress Style; **EDI-IAw**: Interoceptive Awareness Scale of the Eating Disorder Inventory; **IAS**: Interoceptive Accuracy Scale; **IATS**: Interoceptive Attention Scale; **ICQ**: Interoceptive Confusion Questionnaire; **ISQ**: Interoception Sensory Questionnaire; **MAIA**: Multidimensional Assessment of Interoceptive Awareness; **ND**: Noticing; **NW**: Not-Distracting; **NW**: Not-Worrying; **AR**: Attention Regulation; **EA**: Emotional Awareness; **SR**: Self-Regulation; **BL**: Body Listening; **Trusting**: Trusting; **SAQ**: Self-Awareness Questionnaire; **SPI**: Sensory Profile: Interoception; **THISQ**: Three-Domain Interoceptive Sensations Questionnaire.

**Fig 2. Frequency of reported interoceptive self-report scales in the included studies.**

administered this measure [40, 65, 79]. The frequency of reported scales in analyses across these studies is displayed in Fig 2 at the end of this section.

**Body awareness questionnaire (BAQ).** Two studies administered the BAQ [36] to measure' 'interoceptive sensibility' [72, 80]. Both studies analysed BAQ-Total scores.

**Body perception questionnaire (BPQ).** The BPQ [29, 33] was employed by seven studies to measure 'interoception' [59], 'interoceptive awareness' [68], 'interoceptive sensibility' [81–83], 'self-reported interoceptive attention' [40, 65, 66, 77], and 'subjective interoception' [79].

Seven studies used the complete or abbreviated Body Awareness scale (BPQ-BA) [40, 59, 65, 68, 77, 79, 81]. Three studies analysed total scores from the Autonomic Reactivity scale

(BPQ-R-Total), consisting of supra- and sub-diaphragmatic symptoms subscales (BPQ-R-Supra, BPQ-R-Sub, respectively) [68, 77, 79]. Two studies analysed scores for specific BPQ-R-Supra and BPQ-R-Sub scales [65, 79].

**Interoceptive awareness scale of the eating disorder inventory (EDI-IAw).** One study administered the EDI-IAw [84] to measure 'interoceptive awareness' [57]. This study analysed total EDI-IAw scores.

**Interoceptive accuracy scale (IAS).** Six studies administered the IAS [40] to measure 'self-reported interoceptive accuracy' [40, 65, 66, 77, 85], 'interoceptive sensibility' [67], and 'subjective interoception' [79]. IAS-Total scores were analysed, according to results from original and subsequent validation studies [40, 65]. One study included the IAS in their PCA to determine salient components in interoceptive and emotional conceptualisation [67].

**Interoceptive confusion questionnaire (ICQ).** Five studies employed the ICQ [41] to measure 'interoceptive sensibility' [41, 67], 'self-reported interoceptive accuracy' [40, 65] and 'subjective interoception' [79]. All studies analysed and reported on ICQ total scores, according to scoring methods reported by Brewer et al. [41]. One study ran a PCA to determine salient factors in interoceptive and emotional conceptualisation, which included the ICQ [67].

**Interoception sensory questionnaire (ISQ).** Two studies administered the ISQ [43] to measure 'interoceptive confusion' [58] and 'interoceptive challenges' [43]. The ISQ was designed for and validated in adults with ASD. These studies analysed ISQ total scores, based on the factor structure determined by Fiene et al. [43].

**Interoceptive attention scale (IATS).** One study administered the IATS [42] to measure 'self-reported interoceptive attention' and analysed total IATS scores [66].

**Multidimensional assessment of interoceptive awareness (MAIA).** The MAIA was the most frequently employed self-report scale amongst the included studies. Two versions of the MAIA exist: the 32-item version [34] and the revised MAIA-2, consisting of 37-items that improved internal consistency reliability of the scales [35]. The eight scales include Noticing, Not-Distracting (ND), Not-Worrying (NW), Attention Regulation (AR), Emotional Awareness (EA), Self-Regulation (SR), Body Listening (BL), and Trusting. Eighteen studies administered the MAIA or MAIA-2 to assess 'interoceptive awareness' [61, 63, 64, 70, 73–76], 'interoceptive sensibility' [56, 60, 62, 69, 71, 86, 87], and 'subjective interoception' [79].

Seven studies administered translated versions for French [73], Spanish [86], Italian [79], German [71], Greek [62], Portuguese [76], and Taiwanese [87] samples. Despite an English version being validated and published in 2018, two studies involving participants fluent in English employed the 32-item version [64, 69]. Data collection dates were not provided, and it is therefore unclear whether they commenced following publication of the MAIA-2.

Eleven studies reported the eight scales separately [60, 62, 65, 69, 71, 73, 74, 79, 80, 86, 87]. Two studies reported specific MAIA scales (Noticing, AR, EA, BL, Trusting [67]; AR, SR, BL [63]). Six studies computed and reported MAIA total scores, consisting of either the average of scores on the eight scales [60, 79] or summed raw scores [61, 64, 75, 76]. One study conducted multidimensional scaling analyses to produce and score three clusters they labelled 'Attention Regulation' (AR scale), 'Active and Reactive Strategies' (ND, NW, SR, BL scales), and 'Awareness' (Noticing, EA, Trusting scales) [70]. One study ran a PCA to determine salient factors in interoceptive and emotional conceptualisation, which included MAIA scales [67].

**Self-awareness questionnaire (SAQ).** One study developed, validated, and administered the SAQ to measure 'interoceptive awareness' [37]. This study reported on SAQ-Total scores, SAQ-F1 (, and SAQ-F2 scores following factor analysis to determine SAQ factor structure.

**Sensory profile: Interoception (SPI).** One study developed, validated, and administered the SPI to measure 'interoceptive impact' [78]. This study reported on four scales following analysis to determine factor structure: Registration, Avoiding, Sensitivity, Seeking.

**Three-domain interoceptive sensations questionnaire (THISQ).**   One study developed, validated, and administered the THISQ to measure self-reported perception of neutral respiratory, cardiac, and gastroesophageal sensations [45]. The included sensations were designed to differentiate between awareness of neutral sensations and attention to negatively valenced bodily sensations (e.g., dyspnea). This study reported four indices following analyses that determined underlying factor structure: total scores (THISQ-Total), cardiorespiratory activation (THISQ-CRA), cardiorespiratory deactivation (THISQ-CRD), and gastro-esophageal sensations (THISQ-GES).

**Excluded interoceptive self-report scales.**   Two studies administered measures that had not been validated. Brewer et al. [41] administered the State-Emotion Similarity Questionnaire, whereby reported psychometric evidence was unclear; accordingly. these findings were excluded, although the study was retained as they included the ICQ and alexithymia scales. Zamariola et al. [80] reported findings related the Interoceptive Awareness Questionnaire (IAQ). Boegarts et al. [44] recently validated this questionnaire—now named the Interoceptive Sensitivity and Attention Questionnaire. Newer findings indicate a different factor structure to that initially reported [80]. As such, IAQ findings were excluded, but the study was overall retained as they employed the MAIA, BAQ, and alexithymia scales.

**Alexithymia scales.**   Three scales were administered to measure the alexithymia construct: the Bermond-Vorst Alexithymia Questionnaire (BVAQ) [88], Perth Alexithymia Questionnaire (PAQ) [89], and the Toronto Alexithymia Scale, 20-item version (TAS-20) [90].

Most studies administered the TAS-20 ($n$ = 31) to measure alexithymia. The TAS scales include Difficulty Identifying Feelings (DIF) capturing difficulty identifying and distinguishing between feelings and bodily sensations, Difficulty Describing Feelings (DDF) measuring the ability to communicate feelings to other people, and Externally Oriented Thinking (EOT) capturing preferences for focusing on external events rather than inner experiences, with total scores (TAS-Total) reflecting overall alexithymia. Over half of the studies reported findings relevant to TAS-Total only ($n$ = 17, 53.1%). Thirteen studies reported on TAS-Total, DIF, DDF, and EOT. Two studies reported on only facet-level alexithymia using the three TAS-20 scales [45, 67].

Two studies administered the PAQ and utilised PAQ-Total scores to index global alexithymia, consisting of DIF, DDF, and EOT facets [63, 78]. One study administered the BVAQ to measure alexithymia, computing cognitive (BVAQ-C) and affective (BVAQ-A) domains [79]. BVAQ-C measures the degree to which individuals can define arousal states, describe or communicate emotional reactions, and seeking out explanations of emotional reactions, whereas BVAQ-A assesses inclinations to fantasise, imagine, or daydream, and degrees to which someone is emotionally aroused by emotion inducing events.

We investigated whether there was convergent evidence for these alexithymia measures assessing the same construct. Preece et al. [89] reported that TAS-Total and PAQ-Total were strongly correlated ($r$ = 0.76, $p$ < .001). Zahid et al. [63] similarly found a strong correlation between the scales ($r$ = 0.67, $p$ < .001). Gaggero et al. [79] reported significant correlations between and TAS-Total and BVAQ-C ($r$s = 0.68 to 0.85, $p$ < .001). Accordingly, TAS-Total, PAQ-Total, BVAQ-Total, and BVAQ-C scores were considered to assess global alexithymia. Gaggero et al. [79] reported correlations between TAS-20 scores and BVAQ-A, which indicated distinctness ($r$s = 0.07–0.14, $p$s >.05). Accordingly, BVAQ-A findings were not incorporated into this review. Fig 3 displays the frequency of reported alexithymia scales.

## The relationship between subjective interoception and alexithymia

The relationship between subjective interoception and alexithymia was interpreted, according to each interoceptive self-report scale and its relationship with global alexithymia, DIF, DDF,

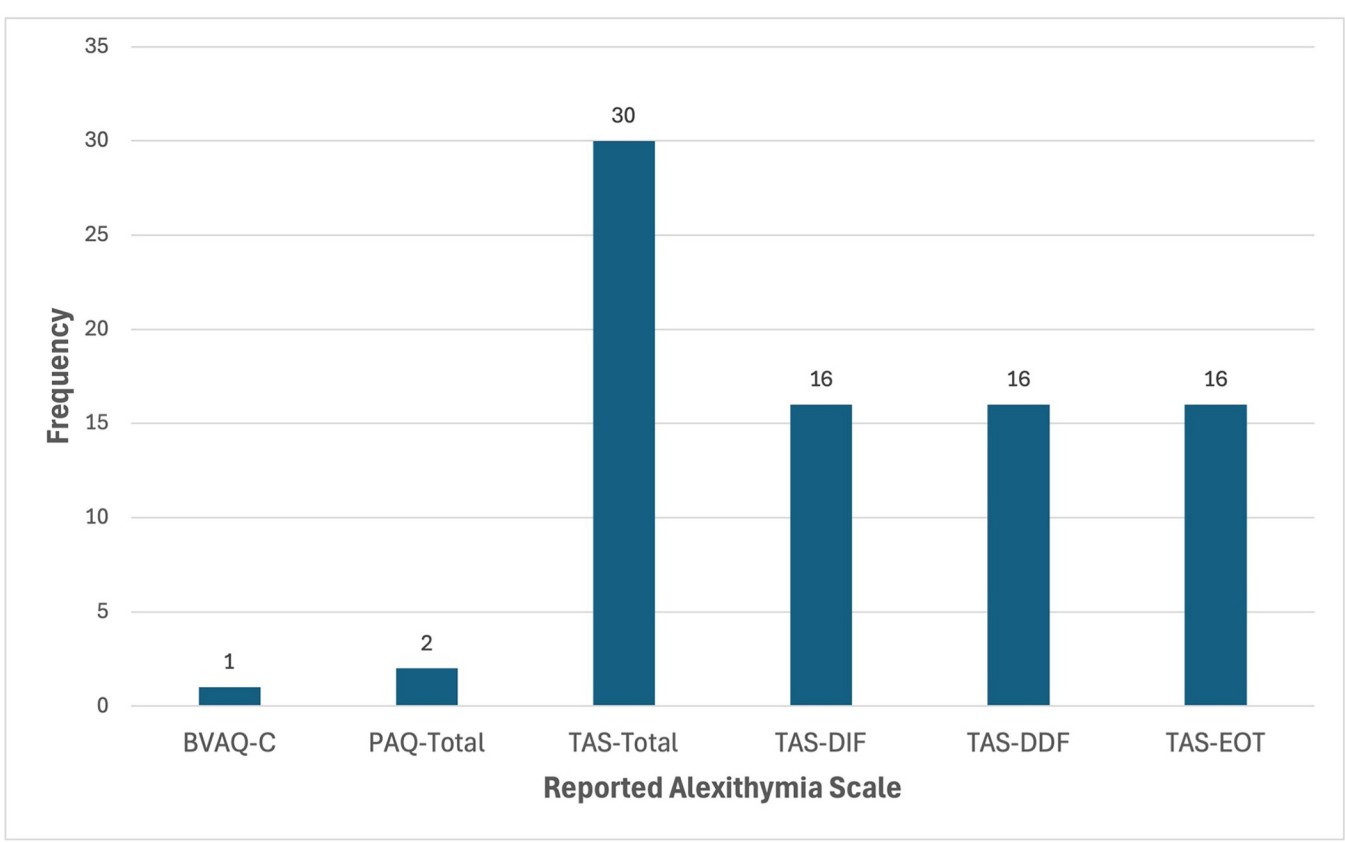

**BVAQ-C**: Bermond-Vorst Alexithymia Questionnaire, Cognitive domain; **PAQ**: Perth Alexithymia Questionnaire; **TAS**: Toronto Alexithymia Scale; **DIF**: Difficulty Identifying Feelings; **DDF**: Difficulty Describing Feelings; **EOT**: Externally Oriented Thinking.

**Fig 3. Frequency of reported alexithymia scales in the included studies.**

and EOT, where $k \geq 2$. Results in the following sections are discussed according to the strength and directionality of these associations. Due to the number of analyses performed, results have been summarised. Subgroup analyses were performed but have been omitted, due to the small sample sizes within many subgroups. An overview of these analyses is provided at https://osf.io/ywg8z. Data and study details are provided in the S6–S9 Files. Full results for primary and secondary meta-analyses and forest and funnel plots are available at https://osf.io/3nsyc/. Fig 4 provides a workflow of the primary and secondary analyses performed.

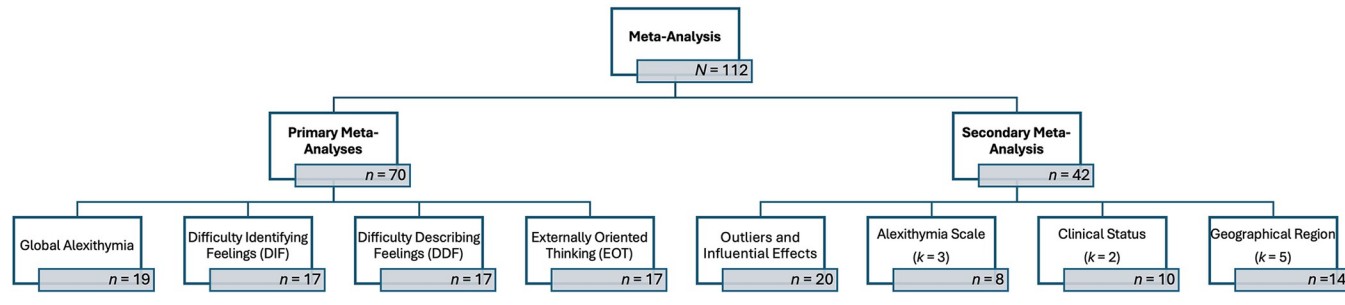

**Fig 4. Workflow of performed meta-analyses.**

**Table 1.** Associations between interoceptive self-report scales and global alexithymia.

| Interoceptive Scale | k | Sample Size | r | z | p | 95% CI Lower | 95% CI Upper | Q | df | p | $I^2$ (%) | Egger's Test t | Egger's Test p | Trim-and-Fill No. Imputed Studies |
|---|---|---|---|---|---|---|---|---|---|---|---|---|---|---|
| BAQ | 2 | 315 | -0.17 | -3.12 | .002 | -0.27 | -0.06 | 0.14 | 1 | .708 | 0.9 | | | |
| BPQ-BA | 13 | 2107 | -0.05 | -0.29 | .181 | -0.12 | 0.09 | 14.31 | 12 | .281 | 66.6 | -0.12 | .906 | 0 |
| BPQ-R-Total | 8 | 1347 | 0.36 | 7.97 | < .001 | 0.28 | 0.44 | 7.77 | 7 | .354 | 59.2 | | | |
| BPQ-R-Supra | 7 | 1167 | 0.34 | 9.23 | < .001 | 0.27 | 0.40 | 8.24 | 6 | .221 | 51.5 | | | |
| BPQ-R-Sub | 7 | 1157 | 0.22 | 7.53 | < .001 | 0.16 | 0.27 | 4.45 | 6 | .617 | 21.9 | | | |
| EDI-Aw | 4 | 312 | 0.26 | 3.90 | < .001 | 0.11 | 0.40 | 4.48 | 3 | .214 | 43.0 | | | |
| IAS | 11 | 2263 | -0.30 | -9.82 | < .001 | -0.35 | -0.24 | 6.13 | 10 | .804 | 44.0 | -2.63 | .027 | 0 |
| IATS | 2 | 581 | 0.22 | 5.55 | < .001 | 0.14 | 0.30 | 0.12 | 1 | .912 | 0.0 | | | |
| ICQ | 9 | 2355 | 0.57 | 12.32 | < .001 | 0.49 | 0.63 | 38.92 | 7 | < .001 | 76.3 | | | |
| ISQ | 2 | 68 | 0.53 | 4.70 | < .001 | 0.33 | 0.71 | 0.11 | 1 | .737 | 0.6 | | | |
| MAIA-AR | 20 | 4279 | -0.26 | -13.05 | < .001 | -0.30 | -0.22 | 16.15 | 19 | .623 | 27.0 | -4.64 | < .001 | 0 |
| MAIA-BL | 19 | 3942 | -0.23 | -9.73 | < .001 | -0.28 | -0.19 | 0.16 | 18 | .674 | 0.2 | -2.22 | .043 | 0 |
| MAIA-EA | 17 | 3183 | -0.18 | -8.11 | < .001 | -0.23 | -0.14 | 0.53 | 16 | .465 | 34.2 | -3.11 | .007 | 1 |
| MAIA-ND | 18 | 3520 | -0.21 | -7.87 | < .001 | -0.26 | -0.16 | 28.77 | 17 | .037 | 51.4 | -3.88 | .001 | 2 |
| MAIA-Noticing | 17 | 3183 | -0.18 | -6.99 | < .001 | -0.23 | -0.13 | 21.89 | 16 | .147 | 45.4 | -2.48 | .025 | 0 |
| MAIA-NW | 17 | 3183 | -0.17 | -5.28 | < .001 | -0.22 | -0.09 | 37.61 | 16 | .002 | 63.6 | -1.29 | .217 | 2 |
| MAIA-SR | 17 | 3183 | -0.28 | -11.34 | < .001 | -0.32 | -0.23 | 20.55 | 16 | .197 | 40.5 | -4.13 | < .001 | 1 |
| MAIA-Trusting | 18 | 3520 | -0.34 | -15.76 | < .001 | -0.38 | -0.30 | 17.17 | 17 | .443 | 33.9 | -6.58 | < .001 | 2 |
| MAIA-Total | 10 | 1488 | -0.41 | -13.10 | < .001 | -0.46 | -0.35 | 10.83 | 9 | .287 | 35.0 | -4.47 | .002 | 4 |

*k*: number of effects; *r*: pooled correlation, CI: confidence interval; BAQ: Body Awareness Questionnaire; BPQ-BA: Body Perception Questionnaire-Body Awareness Scale; BPQ-R-Total: Body Perception Questionnaire-Autonomic Reactivity Scale; BPQ-R-Supra: Body Perception Questionnaire-Supradiaphragmatic Symptoms subscale; BPQ-R-Sub: Body Perception Questionnaire-Subdiaphragmatic Symptoms subscale; EDI-IAw: Interoceptive Awareness subscale of Eating Disorder Inventory; IAS: Interoceptive Accuracy Scale; IATS: Interoceptive Attention Scale; ICQ: Interoceptive Confusion Questionnaire; ISQ: Interoceptive Sensory Questionnaire; MAIA: Multidimensional Assessment of Interoceptive Awareness; MAIA-AR: Attention Regulation subscale; MAIA-BL: Body Listening subscale; MAIA-EA: Emotional Awareness subscale; MAIA-ND: Not-Distracting subscale; MAIA-NW: Not-Worrying subscale; MAIA-SR: Self-Regulation subscale.

**Meta-analyses of interoceptive self-report scales and global alexithymia.** In Table 1, we report the results for each applicable self-report scale in relation to global alexithymia, indexed by TAS-20, PAQ, and BVAQ-C total scores. We observed significant effects for 18 interoceptive scales which ranged from large to small in strength and varied in directionality (see Table 1). Data and study details are reported in the S6 File. Full results and plots are available at https://osf.io/ecjdx.

Meta-analysis demonstrated a strong positive association for the ICQ and alexithymia, $r(9)$ = 0.57, $p < .001$, suggesting greater struggles to interpret non-affective interoceptive states coincides with difficulties identifying and describing feelings, and a preference for externally oriented thinking. A strong positive effect was further observed between the ISQ and alexithymia, $r(2) = 0.53$, $p < .001$, indicating that greater difficulty registering or interpreting interoceptive sensations relates to higher alexithymia. A moderate positive effect was identified for BPQ-R-Total, $r(8) = 0.36$, $p < .001$, suggesting that heightened autonomic stress response activation, experienced as frequent somatic symptoms, coincides with higher alexithymia. This moderate effect extended to BPQ-R-Supra and alexithymia $r(7) = 0.34$, $p < .001$, indicating that more frequent somatic symptoms above the diaphragm (e.g., breathing problems) relates to higher alexithymia. The association was weaker for reactivity for symptoms below the diaphragm (BPQ-R-Sub), $r(7) = 0.22$, $p < .001$. Other weak associations were identified. The

IATS and alexithymia were positively associated, $r(2) = 0.26$, $p < .001$, indicating that greater self-reported attention to internal signals was associated with increased alexithymia. Moreover, a weak positive correlation was found between EDI-IAw and alexithymia, $r(4) = 0.26$, $p < .001$, suggesting that poorer discrimination between sensations and feelings, and hunger and satiety is associated with higher alexithymia.

Contrastingly, we identified a moderate negative association between MAIA-Total scores and alexithymia, $r(10) = -0.41$, $p < .001$, suggesting that adaptive interoception relates to lower alexithymia. We further found that MAIA-Trusting and alexithymia were moderately negatively associated $r(18) = -0.34$, $p < .001$, indicating that experiencing one's body as safe and trustworthy is associated with lower alexithymia. A medium negative correlation between the IAS and alexithymia was also identified, $r(11) = -0.30$, $p < .001$, indicating that stronger perceived capacities for accurately perceiving interoceptive signals is associated with decreased alexithymia. Negative, albeit, weaker associations with alexithymia were identified for MAIA-AR, $r(20) = -0.26$, $p < .001$, and MAIA-SR, $r(17) = -0.28$, $p < .001$, suggesting that stronger abilities for sustaining and controlling attention to body sensations and regulation of distress through bodily attention coincides with lower alexithymia. Weak effects with alexithymia were also identified for MAIA-BL, $r(19) = -0.23$, $p < .001$, and MAIA-ND $r(18) = -0.22$, $p < .001$, indicating more active listening to the body for insight and not ignoring or distracting oneself from pain or discomfort and is associated with decreased alexithymia. Weaker negative associations were shown for alexithymia and MAIA-Noticing, $r(17) = -0.18$, $p < .001$, MAIA-EA, $r(17) = -0.18$, $p < .001$, MAIA-NW, $r(17) = -0.17$, $p < .001$, and the BAQ $r(2) = -0.17$, $p < .001$. Collectively, this suggests that lower alexithymia relates to higher awareness of neutral, comfortable, and uncomfortable body sensations, and the connection between body sensations and emotional states, not worrying or experiencing emotional distress with pain or discomfort, and greater attention to non-affective bodily states. However, we did not identify a significant association between BPQ-BA and alexithymia, $r(13) = -0.05$, $p = .181$.

**Meta-analyses of interoceptive self-report scales and difficulty identifying feelings (DIF).** In Table 2, results for each applicable self-report scale in relation to DIF are reported, showing significant effects for 15 interoceptive scales. For data and study details, see the S7 File. Full results and plots are provided at https://osf.io/6qw7u.

Meta-analysis demonstrated a strong positive pooled correlation for the ICQ and DIF, $r(3) = 0.51$, $p < .001$, suggesting greater interoceptive confusion is associated with difficulties in distinguishing between different emotions and insufficient realisation that physical sensations may be the manifestation of emotions. A moderate positive correlation was further identified for BPQ-R-Total, $r(3) = 0.48$, $p < .001$, suggesting that heightened autonomic stress response activation coincides with greater DIF. This moderate effect was further observed for BPQ-R-Supra, $r(4) = 0.40$, $p < .001$, and BPQ-R-Sub $r(4) = 0.30$, $p < .001$, indicating that frequent experiences of somatic symptoms above and below the diaphragm relates to more pronounced DIF. A positive association was also observed for IATS and DIF, but weaker in strength, $r(2) = 0.26$, $p < .001$, indicating that greater self-reported attention to internal signals is associated with increased DIF.

Conversely, we identified a moderate negative association between MAIA-Total scores and alexithymia, $r(6) = -0.34$, $p < .001$, suggesting that more adaptive interoceptive percepts relate to less DIF. We also found that MAIA-Trusting and alexithymia were moderately negatively associated $r(10) = -0.33$, $p < .001$, indicating that experiencing one's body as safe and trustworthy is associated with less DIF. An overall negative correlation between the IAS and alexithymia was also identified, $r(5) = -0.29$, $p < .001$, indicating that stronger perceived capacities for accurately perceiving interoceptive signals is associated with lower DIF. Negative, albeit weaker, associations with alexithymia were identified for MAIA-NW, $r(9) = -0.27$, $p < .001$,

**Table 2. Associations between interoceptive self-report scales and Difficulty Identifying Feelings (DIF).**

| Interoceptive Scale | k | Sample Size | r | z | p | 95% CI Lower | 95% CI Upper | Q | df | p | $I^2$ (%) | Egger's Test t | Egger's Test p | Trim-and-Fill No. Imputed Studies |
|---|---|---|---|---|---|---|---|---|---|---|---|---|---|---|
| BAQ | 2 | 315 | -0.07 | -1.28 | .201 | -0.17 | 0.03 | 0.03 | 1 | .855 | 0.1 | | | |
| BPQ-BA | 6 | 1478 | 0.00 | -0.06 | .635 | -0.07 | 0.15 | 3.42 | 5 | .635 | 26.3 | | | |
| BPQ-R-Total | 3 | 814 | 0.48 | 10.31 | < .001 | 0.4 | 0.55 | 3.70 | 2 | .157 | 52.3 | | | |
| BPQ-R-Supra | 4 | 1428 | 0.40 | 5.14 | < .001 | 0.25 | 0.52 | 20.52 | 3 | < .001 | 86.5 | | | |
| BPQ-R-Sub | 4 | 1428 | 0.30 | 4.40 | < .001 | 0.17 | 0.43 | 24.69 | 3 | < .001 | 84.8 | | | |
| IAS | 5 | 1633 | -0.29 | -9.99 | < .001 | -0.35 | -0.24 | 3.01 | 4 | .557 | 27.0 | | | |
| IATS | 2 | 581 | 0.27 | 6.87 | < .001 | 0.20 | 0.34 | 0.19 | 1 | .687 | 1.5 | | | |
| ICQ | 3 | 1178 | 0.51 | 18.53 | < .001 | 0.45 | 0.55 | 0.20 | 2 | .907 | 0.05 | | | |
| MAIA-AR | 10 | 2677 | -0.25 | -6.12 | < .001 | -0.32 | -0.17 | 23.10 | 9 | .006 | 72.9 | -3.57 | .007 | 0 |
| MAIA-BL | 11 | 2849 | -0.12 | -2.55 | < .001 | -0.21 | -0.03 | 12.70 | 10 | .241 | 81.8 | -2.55 | .071 | 3 |
| MAIA-EA | 10 | 2833 | -0.08 | -2.91 | .004 | -0.13 | -0.02 | 11.39 | 9 | .250 | 41.7 | -0.62 | .554 | 0 |
| MAIA-ND | 10 | 2599 | -0.17 | -5.22 | < .001 | -0.24 | -0.11 | 10.60 | 9 | .304 | 58.5 | -2.77 | .024 | 0 |
| MAIA-Noticing | 11 | 2849 | -0.11 | -4.07 | < .001 | -0.15 | -0.06 | 12.50 | 10 | .253 | 42.8 | 2.63 | .027 | 2 |
| MAIA-NW | 9 | 2356 | -0.27 | -10.16 | < .001 | -0.32 | -0.22 | 8.60 | 8 | .378 | 32.0 | | | |
| MAIA-SR | 11 | 2849 | -0.26 | -6.15 | < .001 | -0.31 | -0.16 | 22.05 | 10 | .015 | 72.6 | -3.74 | .005 | 0 |
| MAIA-Trusting | 10 | 2619 | -0.33 | -6.41 | < .001 | -0.44 | -0.26 | 22.51 | 9 | .007 | 83.3 | -6.35 | < .001 | 0 |
| MAIA-Total | 6 | 1360 | -0.34 | -11.36 | < .001 | -0.41 | -0.30 | 2.89 | 5 | .716 | 23.9 | | | |

k: number of effects; r: pooled correlation, CI: confidence interval; BAQ: Body Awareness Questionnaire; BPQ-BA: Body Perception Questionnaire-Body Awareness Scale; BPQ-R-Total: Body Perception Questionnaire-Autonomic Reactivity Scale; BPQ-R-Supra: Body Perception Questionnaire-Supradiaphragmatic Symptoms subscale; BPQ-R-Sub: Body Perception Questionnaire-Subdiaphragmatic Symptoms subscale; IAS: Interoceptive Accuracy Scale; IATS: Interoceptive Attention Scale; ICQ: Interoceptive Confusion Questionnaire; MAIA: Multidimensional Assessment of Interoceptive Awareness; MAIA-AR: Attention Regulation subscale; MAIA-BL: Body Listening subscale; MAIA-EA: Emotional Awareness subscale; MAIA-ND: Not-Distracting subscale; MAIA-NW: Not-Worrying subscale; MAIA-SR: Self-Regulation subscale.

MAIA-AR, $r(10)$ = -0.25, $p < .001$, and MAIA-SR, $r(11)$ = -0.26, $p < .001$, suggesting that not worrying or experiencing emotional distress with pain or discomfort, alongside stronger abilities for sustaining and controlling attention to body sensations and regulation of distress through bodily attention coincides with reduced DIF. Weak effects with DIF were further identified for MAIA-ND $r(10)$ = -0.17, $p < .001$, MAIA-BL, $r(11)$ = -0.12, $p < .001$, MAIA-Noticing, $r(11)$ = -0.11, $p < .001$, and MAIA-EA, $r(10)$ = -0.08, $p = .004$. Together, this suggests that less DIF relates to not distracting or ignoring painful or uncomfortable sensations, active listening to the body for insight, awareness of neutral, comfortable, and uncomfortable body sensations, and the connection between body sensations and emotional states, and greater attention to non-affective bodily states. However, DIF was not significantly associated with BPQ-BA, $r(6)$ = 0.00, $p = .635$, or the BAQ, $r(2)$ = - 0.07, $p = .201$.

**Meta-analyses of interoceptive self-report scales and difficulty describing feelings (DDF).** In Table 3, results for each applicable self-report scale in relation to DDF are reported, showing significant effects for 16 interoceptive scales, ranging from moderate to weak in strength and varying in directionality. For data and study details, see the S8 File. Full results and plots are available at https://osf.io/weab3.

Meta-analysis demonstrated a moderate positive association for the ICQ and DDF, $r(4)$ = 0.40, $p < .001$, suggesting greater interoceptive confusion is associated with difficulties in verbally expressing emotions. A moderate positive correlation was further identified for BPQ-R-Supra, $r(4)$ = 0.31, $p < .001$, indicating that frequent experiences of somatic symptoms

**Table 3. Associations between interoceptive self-report scales and Difficulty Describing Feelings (DDF).**

| Interoceptive Scale | k | Sample Size | r | z | p | 95% CI | | Heterogeneity | | | | Publication Bias | | |
|---|---|---|---|---|---|---|---|---|---|---|---|---|---|---|
| | | | | | | Lower | Upper | Q | df | p | $I^2$ (%) | Egger's Test | | Trim-and-Fill |
| | | | | | | | | | | | | t | p | No. Imputed Studies |
| BAQ | 2 | 315 | -0.14 | -2.52 | .012 | -0.25 | 0.03 | 0.0320 | 1 | .571 | 4.2 | | | |
| BPQ-BA | 8 | 2095 | 0.04 | 0.60 | .949 | -0.17 | 0.10 | 22.72 | 7 | .002 | 85.7 | | | |
| BPQ-R-Total | 3 | 814 | 0.25 | 4.24 | < .001 | 0.14 | 0.36 | 6.48 | 2 | .039 | 67.0 | | | |
| BPQ-R-Supra | 4 | 1428 | 0.31 | 5.19 | < .001 | 0.19 | 0.40 | 15.65 | 3 | < .001 | 78.1 | | | |
| BPQ-R-Sub | 4 | 1428 | 0.19 | 4.76 | < .001 | 0.11 | 0.27 | 6.61 | 3 | .085 | 53.2 | | | |
| IAS | 5 | 1633 | -0.21 | -6.17 | < .001 | -0.22 | -0.14 | 5.04 | 4 | .283 | 41.7 | | | |
| IATS | 2 | 581 | 0.19 | 4.65 | < .001 | 0.11 | 0.26 | 0.27 | 1 | .603 | 3.1 | | | |
| ICQ | 4 | 1428 | 0.43 | 8.75 | < .001 | 0.34 | 0.51 | 8.75 | 3 | < .001 | 71.9 | | | |
| MAIA-AR | 11 | 2849 | -0.22 | 9.08 | < .001 | -0.27 | -0.02 | 5.02 | 10 | .890 | 19.6 | -3.04 | .014 | 0 |
| MAIA-BL | 11 | 2849 | -0.19 | 6.31 | < .001 | -0.25 | -0.12 | 16.72 | 10 | .081 | 55.4 | -1.99 | .078 | 1 |
| MAIA-EA | 11 | 2849 | -0.12 | -4.76 | < .001 | -0.17 | -0.07 | 11.76 | 10 | .301 | 40.9 | -1.69 | .140 | 1 |
| MAIA-ND | 11 | 2849 | -0.19 | -5.44 | < .001 | -0.26 | -0.12 | 30.08 | 10 | < .001 | 67.4 | -3.38 | .008 | 4 |
| MAIA-Noticing | 10 | 2833 | -0.10 | -3.17 | .002 | -0.17 | -0.04 | 19.79 | 9 | .009 | 67.3 | -2.08 | .071 | 0 |
| MAIA-NW | 10 | 2543 | -0.14 | -3.22 | < .001 | -0.22 | -0.05 | 12.18 | 8 | .203 | 74.0 | -1.93 | .090 | 1 |
| MAIA-SR | 11 | 2849 | -0.20 | -7.07 | < .001 | -0.26 | -0.15 | 17.99 | 10 | .055 | 51.7 | -3.04 | .014 | 2 |
| MAIA-Trusting | 11 | 2849 | -0.28 | -13.13 | < .001 | -0.32 | -0.24 | 6.56 | 10 | .766 | 18.9 | -4.40 | .002 | 1 |
| MAIA-Total | 6 | 1360 | -0.32 | -10.03 | < .001 | -0.37 | -0.28 | 5.09 | 5 | .405 | 30.0 | | | |

k: number of effects; r: pooled correlation, CI: confidence interval; BAQ: Body Awareness Questionnaire; BPQ-BA: Body Perception Questionnaire-Body Awareness Scale; BPQ-R-Total: Body Perception Questionnaire-Autonomic Reactivity Scale; BPQ-R-Supra: Body Perception Questionnaire-Supradiaphragmatic Symptoms subscale; BPQ-R-Sub: Body Perception Questionnaire-Subdiaphragmatic Symptoms subscale; IAS: Interoceptive Accuracy Scale; IATS: Interoceptive Attention Scale; ICQ: Interoceptive Confusion Questionnaire; MAIA: Multidimensional Assessment of Interoceptive Awareness; MAIA-AR: Attention Regulation subscale; MAIA-BL: Body Listening subscale; MAIA-EA: Emotional Awareness subscale; MAIA-ND: Not-Distracting subscale; MAIA-NW: Not-Worrying subscale; MAIA-SR: Self-Regulation subscale.

above the diaphragm relates to more pronounced DDF. Although they were weaker in magnitude, positive associations were also observed for BPQ-R-Total, $r(3) = 0.25$, $p < .001$, BPQ-R-Sub $r(4) = 0.19$, $p < .001$, and IATS, $r(2) = 0.21$, $p < .001$, suggesting that heightened autonomic stress response activation, frequently felt somatic symptoms below the diaphragm, and greater self-reported attention to internal signals relate to increased DDF. Although positive in direction, we found that DDF was not significantly associated with BPQ-BA, $r(8) = 0.04$, $p = .949$.

By contrast, a moderate negative association between MAIA-Total scores and DDF was identified, $r(6) = -0.32$, $p < .001$, indicating that more adaptive interoceptive percepts relate to lower DDF. DDF was further found to be negatively associated with MAIA-Trusting $r(11) = -0.28$, $p < .001$, MAIA-AR, $r(11) = -0.22$, $p < .001$, MAIA-SR, $r(10) = -0.20$, $p < .001$, and the IAS, $r(10) = -0.21$, $p < .001$. Together, this suggests experiencing one's body as safe and trustworthy, stronger abilities for sustaining and controlling attention to body sensations and regulation of distress through bodily attention, and stronger perceived capacities for accurately detecting interoceptive signals coincides with reduced DDF. Negative associations were also identified between DDF and MAIA-BL $r(11) = -0.19$, MAIA-ND, $r(11) = -0.19$, $p < .001$. This indicates that lower DDF is related to active listening to the body for insight, and not ignoring or distracting oneself from painful or uncomfortable sensations. The weakest negative effects were shown for MAIA-Noticing $r(11) = -0.10$, MAIA-EA, $r(11) = -0.12$, MAIA-NW, $r(10) = -0.14$, and the BAQ $r(2) = -0.14$. Collectively, this suggests that awareness of neutral,

**Table 4. Associations between interoceptive self-report scales and Externally Oriented Thinking (EOT).**

| Interoceptive Scale | k | Sample Size | r | z | p | 95% CI Lower | 95% CI Upper | Q | df | p | $I^2$ (%) | Egger's Test t | Egger's Test p | Trim-and-Fill No. Imputed Studies |
|---|---|---|---|---|---|---|---|---|---|---|---|---|---|---|
| BAQ | 2 | 315 | -0.19 | -3.31 | .012 | -0.30 | -0.08 | 0.57 | 1 | .449 | 11.4 | | | |
| BPQ-BA | 8 | 2095 | 0.02 | 0.25 | .352 | -0.12 | 0.15 | 26.14 | 7 | < .001 | 84.9 | | | |
| BPQ-R-Total | 2 | 489 | 0.17 | 3.83 | .048 | 0.00 | 0.32 | 3.83 | 2 | .050 | 80.7 | | | |
| BPQ-R-Supra | 3 | 1103 | 0.14 | 2.20 | .028 | 0.01 | 0.26 | 7.53 | 2 | .023 | 74.2 | | | |
| BPQ-R-Sub | 4 | 1428 | 0.05 | 1.03 | .116 | -0.04 | 0.15 | 6.72 | 3 | .081 | 60.3 | | | |
| IAS | 5 | 1633 | -0.16 | 5.2 | < .001 | -0.21 | -0.1 | 3.17 | 4 | .530 | 25.8 | | | |
| IATS | 2 | 581 | 0.03 | 0.59 | .555 | -0.16 | 0.11 | 0.49 | 1 | .484 | 8.8 | | | |
| ICQ | 4 | 1428 | 0.24 | 4.24 | < .001 | 0.13 | 0.34 | 10.79 | 3 | < .001 | 76.1 | | | |
| MAIA-AR | 11 | 2849 | -0.21 | -7.58 | < .001 | -0.26 | -0.16 | 18.56 | 10 | .046 | 47.2 | -3.04 | .014 | 0 |
| MAIA-BL | 10 | 2543 | -0.21 | -6.03 | < .001 | -0.28 | -0.15 | 10.26 | 9 | .330 | 35.5 | -3.10 | .013 | 4 |
| MAIA-EA | 10 | 2619 | -0.25 | -8.16 | < .001 | -0.31 | -0.19 | 18.65 | 9 | .028 | 53.6 | -3.96 | .004 | 3 |
| MAIA-ND | 11 | 2849 | -0.13 | -3.44 | < .001 | -0.20 | -0.05 | 23.31 | 10 | .010 | 69.8 | -3.27 | .010 | 2 |
| MAIA-Noticing | 10 | 2541 | -0.14 | -3.63 | < .001 | -0.21 | -0.06 | 30.88 | 9 | < .001 | 67.8 | -1.25 | .247 | 3 |
| MAIA-NW | 10 | 2619 | -0.02 | -0.80 | .423 | -0.07 | 0.03 | 11.98 | 9 | .215 | 37.1 | 0.33 | .749 | 3 |
| MAIA-SR | 11 | 2849 | -0.21 | -6.26 | < .001 | -0.27 | -0.14 | 26.53 | 10 | .003 | 62.4 | -3.08 | .013 | 2 |
| MAIA-Trusting | 10 | 2619 | -0.20 | -6.34 | < .001 | -0.26 | -0.14 | 6.67 | 9 | .671 | 53.4 | -2.39 | .044 | 0 |
| MAIA-Total | 5 | 1052 | -0.24 | -7.02 | < .001 | -0.31 | -0.18 | 3.40 | 4 | .498 | 21.8 | | | |

k: number of effects; r: pooled correlation, CI: confidence interval; BAQ: Body Awareness Questionnaire; BPQ-BA: Body Perception Questionnaire-Body Awareness Scale; BPQ-R-Total: Body Perception Questionnaire-Autonomic Reactivity Scale; BPQ-R-Supra: Body Perception Questionnaire-Supradiaphragmatic Symptoms subscale; BPQ-R-Sub: Body Perception Questionnaire-Subdiaphragmatic Symptoms subscale; IAS: Interoceptive Accuracy Scale; IATS: Interoceptive Attention Scale; ICQ: Interoceptive Confusion Questionnaire; MAIA: Multidimensional Assessment of Interoceptive Awareness; MAIA-AR: Attention Regulation subscale; MAIA-BL: Body Listening subscale; MAIA-EA: Emotional Awareness subscale; MAIA-ND: Not-Distracting subscale; MAIA-NW: Not-Worrying subscale; MAIA-SR: Self-Regulation subscale.

comfortable, and uncomfortable sensations, alongside recognition of the link between sensations and emotions, not experiencing emotional distress with pain or discomfort, and greater attention to non-affective bodily states coincides with lower DDF.

**Meta-analyses of interoceptive self-report scales and externally oriented thinking (EOT).** As shown in Table 4, we observed significant small overall effects for the association between EOT and 13 interoceptive scales. For data and study details, see the S9 File. Full results and plots are available at https://osf.io/2e5mn.

Meta-analysis demonstrated relatively weak positive associations between EOT and the ICQ, $r(4) = 0.24$, $p < .001$, BPQ-R-Total, $r(2) = 0.17$, $p < .001$, and BPQ-R-Supra, $r(3) = 0.14$, $p < .001$. Together, this indicates that greater interoceptive confusion and heightened autonomic stress response activation, particularly experienced as somatic symptoms above the diaphragm, coincides with preferences for focussing on the external environment and hardly on inner experience. We did not find evidence for significant associations between EOT and BPQ-BA, $r(8) = 0.02$, $p = .352$, BPQ-R-Sub, $r(4) = 0.05$, $p = .116$, or the IATS, $r(2) = 0.03$, $p = .555$.

Conversely, weak negative associations between EOT and most MAIA scales and summary scores were identified, including MAIA-total, $r(5) = -0.24$, $p < .001$, MAIA-EA, $r(10) = -0.25$, $p < .001$, MAIA-BL, $r(10) = -0.21$, $p < .001$, MAIA-AR, $r(11) = -0.21$ $p < .001$, MAIA-SR, $r(11) = -0.21$, $p < .001$, MAIA-Trusting, $r(10) = -0.20$, $p < .001$, MAIA-Noticing, $r(10) = -0.14$, $p < .001$, and MAIA-ND, $r(11) = -0.13$, $p < .001$. The BAQ was also positively associated with

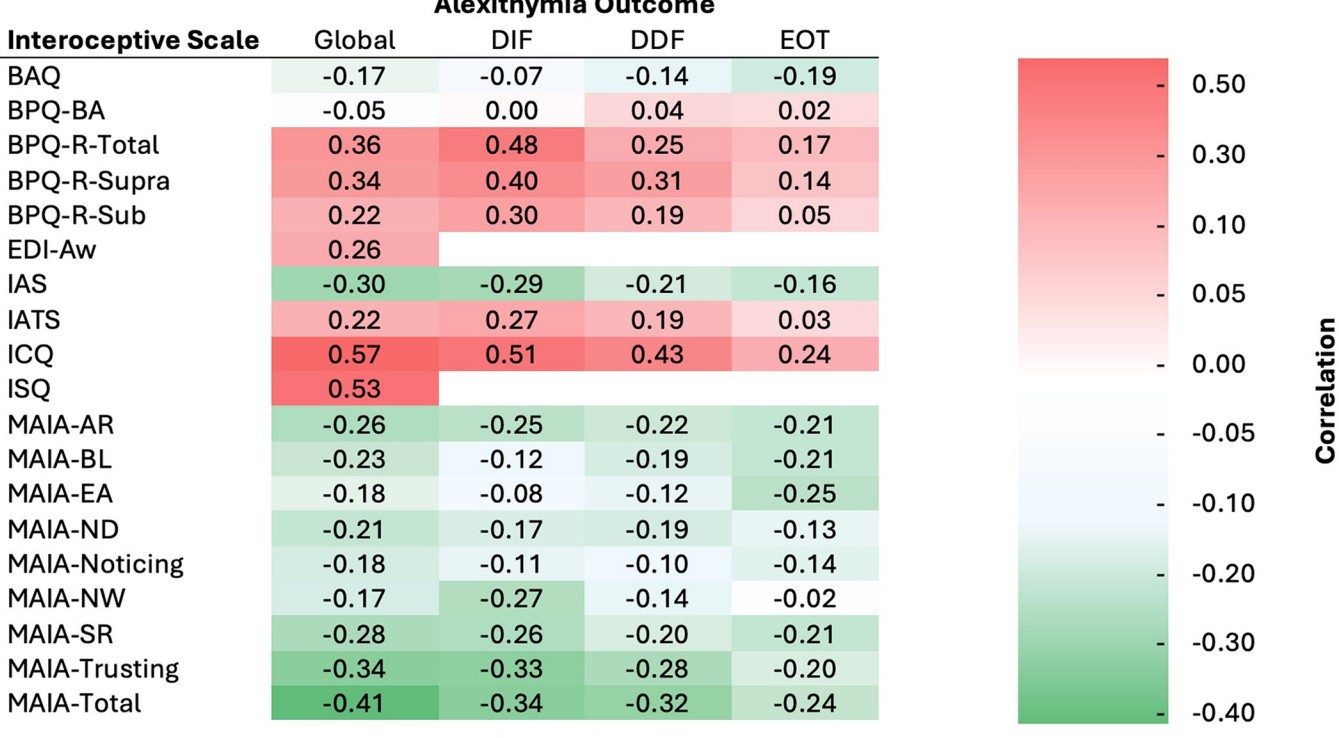

| | Alexithymia Outcome | | | |
| Interoceptive Scale | Global | DIF | DDF | EOT |
| --- | --- | --- | --- | --- |
| BAQ | -0.17 | -0.07 | -0.14 | -0.19 |
| BPQ-BA | -0.05 | 0.00 | 0.04 | 0.02 |
| BPQ-R-Total | 0.36 | 0.48 | 0.25 | 0.17 |
| BPQ-R-Supra | 0.34 | 0.40 | 0.31 | 0.14 |
| BPQ-R-Sub | 0.22 | 0.30 | 0.19 | 0.05 |
| EDI-Aw | 0.26 | | | |
| IAS | -0.30 | -0.29 | -0.21 | -0.16 |
| IATS | 0.22 | 0.27 | 0.19 | 0.03 |
| ICQ | 0.57 | 0.51 | 0.43 | 0.24 |
| ISQ | 0.53 | | | |
| MAIA-AR | -0.26 | -0.25 | -0.22 | -0.21 |
| MAIA-BL | -0.23 | -0.12 | -0.19 | -0.21 |
| MAIA-EA | -0.18 | -0.08 | -0.12 | -0.25 |
| MAIA-ND | -0.21 | -0.17 | -0.19 | -0.13 |
| MAIA-Noticing | -0.18 | -0.11 | -0.10 | -0.14 |
| MAIA-NW | -0.17 | -0.27 | -0.14 | -0.02 |
| MAIA-SR | -0.28 | -0.26 | -0.20 | -0.21 |
| MAIA-Trusting | -0.34 | -0.33 | -0.28 | -0.20 |
| MAIA-Total | -0.41 | -0.34 | -0.32 | -0.24 |

**Correlation**

DIF: Difficulty Identifying Feelings; DDF: Difficulty Describing Feelings; EOT: Externally Oriented Thinking; BAQ: Body Awareness Questionnaire; BPQ-BA: Body Perception Questionnaire-Body Awareness Scale; BPQ-R-Total: Body Perception Questionnaire-Autonomic Reactivity Scale; BPQ-R-Supra: Body Perception Questionnaire-Supradiaphragmatic Symptoms subscale; BPQ-R-Sub: Body Perception Questionnaire-Subdiaphragmatic Symptoms subscale; EDI-IAw: Interoceptive Awareness subscale of Eating Disorder Inventory; IAS: Interoceptive Accuracy Scale; IATS: Interoceptive Attention Scale; ICQ: Interoceptive Confusion Questionnaire; ISQ: Interoception Sensory Questionnaire; MAIA: Multidimensional Assessment of Interoceptive Awareness; MAIA-AR: Attention Regulation subscale; MAIA-BL: Body Listening subscale; MAIA-EA: Emotional Awareness subscale; MAIA-ND: Not-Distracting subscale; MAIA-NW: Not-Worrying subscale; MAIA-SR: Self-Regulation subscale.

**Fig 5. Heat map of pooled correlations between alexithymia domains and interoceptive measures.** The colour gradient illustrates the strength and direction of each pooled correlation, with red indicating positive correlations and green representing negative correlations.

EOT, $r(2)$ = -0.19, $p$ = .012. This suggests that more adaptive interoceptive percepts, including awareness of the connection between sensations and emotions, active listening to the body for insight, and sustaining and controlling attention to body sensations are related to less EOT and more focus on inner experiences. However, we found that MAIA-NW and EOT were not significantly associated, $r(10)$ = -0.02, $p$ = .423.

A heatmap illustrating the pooled correlations between all alexithymic outcomes and interoceptive measures is provided in Fig 5.

**Narrative synthesis for other interoceptive self-reports and alexithymia.** This section reports a narrative synthesis of findings that were not included in meta-analyses. Ernst et al. [68] examined the relationship between BPQ stress reactivity and stress style scales (BPQ-SR, BPQ-SS, respectively) and alexithymia in 18 healthy Swiss adults. They found strong positive correlations for TAS-Total with BPQ-SR, $r(16)$ = 0.73, $p < .05$, and BPQ-SS, $r(16)$ = 0.65, $p < .05$.

Mul et al. [70] presented results relevant to the relationship between alternative MAIA factor structures and global alexithymia, measured by TAS-Total. In a pooled ASD and healthy control sample, TAS-Total strongly negatively correlated with Awareness (Noticing, EA, Trusting), $r(50)$ = -0.57, $p < .001$, and Active and Reactive Strategies factors (ND, NW, SR,

BL), $r(50)$ = -0.57, $p < .001$. The study further found that adults with ASD and alexithymia reported significantly lower scores for MAIA factors than adults with ASD and no alexithymia and the control group.

Ventura-Bort et al. [67] employed a two-factor solution for a PCA to ensure that extracted components reflected general constructs underlying their self-report variables. They identified a Sensibility component—interpreted as reflecting beliefs about the accuracy in detecting internal physiological and emotional states—on which DIF, DDF, and EOT from the TAS-20 all loaded (-0.89, -0.67, -0.39, respectively). MAIA-AR (0.63) MAIA-Trusting (0.61), IAS (0.51), and ICQ (-0.63) loaded onto the Sensibility factor with the TAS-20 scales. A Monitoring component—interpreted as a tendency to focus on internal states—was also identified. Noticing (0.54) and EA (0.72) scales loaded onto this component, indicating heterogeneity to these MAIA scales and alexithymic traits. MAIA-BL was found to cross-load (0.47 loading on both components).

Longarzo et al. [37] examined how SAQ-Total, SAQ-F1 (, and SAQ-F2 related to TAS-Total, DIF, DDF, and EOT TAS-20 scales in healthy university students and staff. Regarding TAS-Total, significant negative correlations were found with SAQ-Total, $r(248)$ = 0.47, $p < .01$, and SAQ-F2, $r(248)$ = 0.37, $p < .01$; the correlation was small with SAQ-F1 $r(248)$ = 0.25, $p < .01$. For DIF, medium positive correlations were shown for SAQ-Total, $r(248)$ = 0.47, $p < .01$, SAQ-F1, $r(248)$ = 0.40, $p < .01$, and SAQ-F2, $r(248)$ = 0.25, $p < .01$. For DDF, a medium positive correlation was reported for SAQ-F2, $r(248)$ = 0.33, $p < .01$; small correlations were found for SAQ-Total $r(248)$ = 0.42, $p < .01$, and SAQ-F1, $r(248)$ = 0.20, $p < .01$. For EOT, there was no correlation with any SAQ score. SAQ-Total significantly positively predicted TAS-Total, explaining 13% of variance ($\beta$ = 0.37, $p < .001$).

Dunn et al. [78] presented findings relevant to SPI scales (Avoidance, Registration, Seeking, Sensitivity) and alexithymia, as measured by PAQ-Total. In university students, a small positive correlation was shown for SPI-Registration and PAQ-Total, $r(72)$ = 0.26, $p < .05$. No correlation was reported for SPI-Avoidance, SPI-Seeking, and SPI-Sensitivity.

Vlemincx et al. [45] presented data relevant to THISQ scales—cardiorespiratory activation (CRA), cardiorespiratory deactivation (CRD), gastro-esophaegeal sensations (GES), THISQ-Total—and alexithymia, as measured by DIF, DDF, and EOT TAS-20 scales. In Dutch- and English-speaking samples, a small positive correlation was found for DIF and THISQ-CRA, $r(729)$ = 0.10, $p < .01$, but this was not significant for THISQ-CRD, THISQ-GES, and THISQ-Total. For DDF, small positive correlations were shown for THISQ-CRA, $r(729)$ = 0.13, $p < .001$, and THISQ-GES, $r(729)$ = 0.09, $p < .05$. No correlation was found for THISQ-CRD and THISQ-Total. For EOT, small negative correlations were found with THISQ-CRA, $r(729)$ = -0.13, $p < .01$, THISQ-CRD, $r(729)$ = -0.09, $p < .05$, THISQ-GES, $r(729)$ = -0.12, $p < .01$, and THISQ-Total, $r(729)$ = -0.14, $p < .001$.

## Discussion

We conducted a systematic review and meta-analysis of the relationship between subjective interoception, as measured by various self-report scales, and alexithymia. The studies were cross-sectional and, using meta-analyses, we determined that an empirical relationship exists between these constructs, contingent upon both the measurement of self-reported interoception and distinct alexithymic facets. Overall, primary meta-analyses indicated that alexithymia is significantly positively associated with measures assessing interoceptive confusion, interoceptive attention, and autonomic stress responses and negatively associated with measures of interoceptive accuracy and adaptive interoception. These relationships were observed to be stronger with difficulty identifying (DIF) and describing feelings (DDF), relative to externally

oriented thinking (EOT). Notably, the BPQ-BA was not significantly associated with any alex-ithymic outcome. Interoceptive research is fraught with issues pertaining to construct defini-tion and operationalisation. Although behavioural paradigms probing objective performance have received attention, recent scrutiny extends to interoceptive self-report scales [26, 31, 32, 39]. Our findings add weight to current contentions advocating for the application of specific interoceptive constructs gauged via questionnaire and provide a basis for understanding the implications of which measures may be suitable for exploring relationships with alexithymia.

## The relationship between interoceptive self-report scales and alexithymia

With respect to global alexithymia, meta-analyses revealed that poorer detection and differen-tiation of internal bodily sensations coincided with higher alexithymia, involving increased dif-ficulties distinguishing between emotions, realising that physical sensations may represent emotions, and articulating emotional experiences. This follows the observed positive associa-tions between alexithymia and interoceptive self-report scales measuring interoceptive confu-sion and awareness in both clinical and non-clinical samples—specifically, the ICQ, ISQ, and EDI-IAw—previously classified as interoceptive accuracy measures in the literature [12, 25, 40]. Conversely, stronger beliefs in accurately perceiving various interoceptive signals (IAS) related to lower levels of alexithymia. Previous meta-analyses have found no significant empir-ical association between task-based interoceptive accuracy and alexithymia [18, 25]. Consistent with Trevisan et al. [25], these findings underscore the importance of subjective trait-based interoceptive accuracy in emotional awareness, suggesting that alexithymia possibly stems from beliefs that interoceptive information is unreliable [22]. Alexithymia is associated with perceived difficulties in conceptualising physiological arousal as representative of emotions (e.g., recognition that elevated heartrate may relate to surprise), and in discriminating between indicators of affective arousal, resulting in less precise distinctions among emotional states and diminished capacities for verbal expression of emotions [25, 91]. Accurate detection of body sensations may facilitate more granular detection, articulation, and conceptualisation of spe-cific emotional states, promoting adaptivity and psychological wellbeing [67].

Higher alexithymia was also associated with heightened autonomic stress response activa-tion and reactivity (BPQ-R-Total), expressed as heightened perceptions of supra- (BPQ-R-Su-pra) and subdiaphragmatic (BPQ-R-Sub) symptoms, and attention to body sensations (IATS). This suggests that increased attention to bodily sensations, suggestive of hypervigilance and homeostatic disturbance, and autonomic nervous system (ANS) activation co-occur with higher global alexithymia. Atypical brain-body communications and the detection of symp-toms implicitly involves interoception. However, it is possible that the BPQ-Reactivity scales are not a strict measure of self-reported interoception, but rather an indication of stress reac-tivity, which can explain the significant associations with alexithymia. Stress involves the inter-play of cognitive (e.g., attentional deployment), affective (e.g., worry, anxiety), and physiological changes (e.g., ANS and hypothalamic-pituitary-adrenocortical [HPA] axis acti-vation) that can manifest as somatic symptoms [92]. Such symptoms may be misinterpreted as indicators of illness requiring medical attention if dissociated from emotional representations [4]. Hypervigilant attention to symptoms, emotion processing deficits, somatisation, and excessive HPA axis activation may perpetuate dysfunctional regulation of bodily systems and interoceptive signal processing, contributing to the onset and maintenance of somatic symp-toms and body-related mental disorders (e.g., somatic symptom disorder) [6, 24, 92, 93]. Nonetheless, our results did not support a positive association between BPQ-BA and alexithy-mia, contrary to previous findings [25]. Consistent with Gaggero et al. [79], there is a stronger association with increased symptomatology reflecting ANS dysfunction rather than the

proposed attentional processes captured by the BPQ-BA. ANS dysfunction is not captured by the BPQ-BA, which has been criticised for confusing item wording, focus on unpleasant sensations, and lack of consideration of how respondents who do not experience such sensations might answer such items [16, 79].

Conversely, lower alexithymia was associated with higher adaptive interoceptive beliefs (MAIA-Total), including interoceptive trusting (MAIA-Trusting), self-regulation (MAIA-SR), attention regulation (MAIA-AR), body listening (MAIA-BL), not-distracting (MAIA-ND), emotional awareness (MAIA-EA), noticing (MAIA-Noticing), not-worrying (MAIA-NW), and self-reported attentiveness to normal non-emotive body processes, including anticipation of bodily reactions (BAQ). These findings complement and extend on the previous meta-analytic results of Trevisan et al. [25]. Whilst we also identified that higher MAIA-Noticing and MAIA-EA related to lower alexithymia, results indicated that global alexithymia was more strongly associated with other adaptive constructs measured by the MAIA. These constructs included mindfully sustaining attention to bodily sensations (MAIA-AR), regulating emotions through attention to sensations (MAIA-SR), active listening to bodily sensations for insight (MAIA-BL), tendencies to not ignore (MAIA-ND) or worry about painful or uncomfortable sensations (MAIA-NW), and deeming bodily sensations as trustworthy (MAIA-Trusting). These are characteristics indicating that sensations are important to the individual [16]. Such adaptive beliefs may cumulatively act as top-down modulators of interoceptive attention [94] and promote greater incorporation of physiology into ongoing emotional experiences [95], thus reducing emotional deficits, such as those typifying alexithymia. On the other hand, these findings indicate that lower adaptive beliefs, indexed by lower MAIA scale scores, can be understood as associated with higher alexithymia.

Data amenable to meta-analysis revealed that higher DIF coincided with greater interoceptive confusion (ICQ), attention (IATS), and autonomic stress response activation (BPQ-R-Total, BPQ-R-Supra, BPQ-R-Sub). Conversely, lower DIF was associated with stronger perceptions of interoceptive accuracy (IAS) and adaptive aspects of subjective interoception (MAIA-Total, MAIA-Trusting, MAIA-NW, MAIA-AR, MAIA-SR, MAIA-ND, MAIA-BL, MAIA-Noticing, and MAIA-EA). These results largely reflected the patterns observed for global alexithymia. However, BPQ-BA and BAQ did not show significant relationships with DIF. Among the facets of alexithymia, DIF notably exhibited the strongest associations with interoceptive self-report scales, which could be due to specific aspects of scale construction or to more general features of emotion generation processes. For instance, during development of the TAS, four items were taken from the EDI-IAw to reflect a domain entailing difficulty identifying and distinguishing between feelings and bodily sensations [96]. Subsequent to revisions, this domain is no longer included in the scale [90]. However, two items are retained in the DIF scale, indicating some conceptual overlap. Additionally, theories of emotion propose that the interpretation of physiological sensations and changes is an essential constituent of affect (valence and arousal), which contributes to the experience of emotions [97–99]. In this view, identification of feelings necessitates adequate allocation of attention to and accuracy in detecting interoceptive signals as a basis for an emotional experience.

With respect to DDF, greater interoceptive confusion (ICQ), stronger autonomic nervous system activation (BPQ-R-Supra, BPQ-R-Total, BPQ-R-Sub), and higher interoceptive attention (IATS) was associated with higher DDF. Conversely, lower DDF coincided with stronger perceptions of interoceptive accuracy (IAS) and adaptive aspects of subjective interoception (MAIA-Total, MAIA-Trusting, MAIA-AR, MAIA-SR, IAS, MAIA-BL, MAIA-ND, MAIA-Noticing, MAIA-EA, MAIA-NW, and BAQ). However, DDF was not significantly associated with BPQ-BA. These results were consistent with the patterns observed for global alexithymia and DIF. Although effect sizes were similar, they were somewhat weaker than those observed

for DIF. Various theories of emotion highlight the involvement of language in shaping and refining emotion concepts and schemas, which enables individuals to recognise emotions as they arise and to effectively communicate them to others [100, 101]. This implies that language is crucial for identifying and ascribing a label to a feeling, promoting effective communication of emotions. Interoceptive beliefs and values may further contribute to the interpretation and incorporation of physiology into ongoing emotional experiences [95]. Emotion categories combining interoceptive sensations and emotional ascriptions may subsequently facilitate the development of more granular categories and concepts that are readily communicable. Considering DIF and DDF together, interoceptive deficits may manifest as general confusion between bodily and affective states [41], which most profoundly affects capacities for identifying and describing feelings.

In relation to EOT, meta-analyses revealed similar overall patterns, albeit with weaker associations compared to DIF and DDF. Such findings support the contention that interoceptive deficits are most strongly linked with deficits in identifying and describing emotions [79]. The observed associations between interoceptive self-report scales and EOT were either small in magnitude or not significant (BPQ-BA, BPQ-R-Sub, IATS, MAIA-NW). Unlike DIF and DDF, which explicitly pertain to emotions, EOT specifically represents a cognitive mode of thinking and appears a distinct factor in alexithymia, which might better reflect cognitive deficits in attending to emotionally relevant stimuli [3] rather than interoceptive sensations.

## Key differences in interoceptive self-report scales

Distinctions between adaptive and maladaptive interoceptive attentional dispositions have been identified using existing measures (i.e., BPQ, MAIA), reflecting maladaptive and adaptive attentional styles [11]. Trevisan et al. [25] quantified these distinctions through meta-analysis and subsequently linked adaptive and maladaptive interoceptive attention to clinical outcomes, including somatisation and alexithymia [12]. Alexithymia, characterised as a marker of atypical interoception [41], therefore serves as a relevant construct for distinguishing between interoceptive self-report scales. In particular, the framework introduced by Desmedt et al. [31] provides a useful structure for classifying interoceptive self-report scales in accordance with four interoceptive factors: sensing, attention, interpretation, and memory.

As adaptive and maladaptive interoceptive processes may promote or hinder effective physiological regulation, maladaptive and adaptive perceptions of interoceptive sensing can be operationalised using existing self-reports. As with attention [11], differentiation is equally important for discerning whether individuals can accurately detect, localise, and discriminate between sensations in ways that support or hinder the maintenance of bodily functioning. With respect to maladaptive interoceptive accuracy, which may impede bodily regulation due to confusion with and poor discrimination of bodily signals, the ICQ, ISQ, and EDI-IAw may serve as suitable measures of maladaptive interoceptive sensing, given their positive relationships with global alexithymia, DIF and DDF, alongside weak associations with adaptive interoceptive scales [43, 65].

Within the suite of self-report scales, adaptive interoceptive sensing is proposed to be encompassed by several measures. Although the IAS has been described as capturing neither adaptive nor maladaptive accuracy [12], our evidence indicates that heightened perceived accuracy in detecting various interoceptive sensations, as indicated by high IAS scores, may mitigate physiological dysregulation that is observed in alexithymia [7]. Accordingly, the IAS could serve as a measure of adaptive interoceptive sensing. Moreover, MAIA-Noticing could also be classified as a measure of adaptive sensing, following the observed associations with alexithymic outcomes. We acknowledge that this reclassification deviates from the scale's

traditional classification as a measure of interoceptive attention [e.g., 11, 12, 102]. However, items reflect the basic, non-emotive detection of interoceptive sensations without necessarily attributing meaning or significance to them, nor explicitly engaging attentional mechanisms.

Additionally, the BAQ potentially captures adaptive interoceptive sensing, as items are heavily weighted toward the accurate detection and discrimination of bodily sensations and states. Accurate sensing is imperative for adaptivity, efficient physiological regulation, and wellbeing [12, 13]. BAQ items arguably represent the inverse of poorer capacities measured by ICQ, ISQ, and EDI-IAw—measures involving confusion and interoceptive inaccuracy beliefs, which may propagate chronic physiological dysregulation. Moreover, the prediction of body reactions to internal and external factors (e.g., energy levels, illness) is also assessed [26, 36]. This possibly involves interoceptive memory, as memory of information about factors that have previously affected physiological regulation forms a constituent in predictive models of interoception [13], involving consolidation and encoding of past experiences to inform anticipation of future physiological responses. We recognise that these classifications stand in contrast to the authors' characterisation of the measure as involving self-reported attentiveness to normal non-emotional body processes [36]. Such subjective beliefs may implicitly involve attentional processes, but arguably represent distinct processes. Moreover, although speculated to tap into somatisation [12], previous findings indicate moderate associations between the BAQ and adaptive interoceptive processes captured by the MAIA [26, 32, 38, 39]. Meta-analysis showed negative overall correlations between total BAQ scores and alexithymia in non-clinical student samples. Despite the omission of the Ricciardi et al. [72] finding in a pooled sample of controls and FMD patients, hypotheses regarding the maintenance of functional neurological symptoms suggest a role for somatisation [103]. They reported that FMD patients had significantly lower BAQ scores compared to controls. If the BAQ was assessing maladaptive interoception, high BAQ scores and positive associations with alexithymia would be plausible in this clinical sample. Considering previous evidence and limited data, we speculate that the BAQ scales measure both adaptive interoceptive sensing and memory.

The THISQ, a relatively new interoceptive scale, demonstrated overall weak associations with alexithymic facets; yet, shows mostly moderate correlations with other measures of adaptive non-emotive sensing, including the BAQ and MAIA-Noticing [45]. It is therefore proposed that the THISQ is potentially a measure of adaptive interoceptive sensing of sensations at rest (e.g., typical breathing) and following exertion (e.g., dyspnea).

The present evidence further supports previous assertions regarding the role of MAIA factors in facilitating adaptive attentional processes centred in interoceptive processing [11, 12]. Salient aspects assessable via self-report include not-distracting (MAIA-ND), attention regulation (MAIA-AR), and self-regulation (MAIA-SR). Together, these facets involve a lack of ignoring and acceptance of noxious, uncomfortable sensations, attentional control toward bodily sensations, and regulation of distress through attentional control. These mechanisms are inversely related to alexithymia, underscoring the importance of adaptive interoceptive attentional and regulatory beliefs and mechanisms for emotion identification and articulation.

In concordance with the observed lack of association with alexithymia, we cannot conclusively suggest employment of the BPQ-BA scale, despite previous recommendations [28, 40] and its description as a key measure of maladaptive interoceptive attention [12]. Although limited findings were presented for the IATS [66], the measure was specifically developed to capture subjective interoceptive attention, and may tap into aspects of bodily hypervigilance and homeostatic disturbance based on observed relationships with alexithymia. This is supported by recent experience-sampling findings indicating that higher IATS scores in daily life are associated with more negative valence and fatigue [104]. Whilst IATS and BPQ-BA scores are

positively related [42, 66], IATS items are phrased to gauge whether attention is focused on various interoceptive sensations most of the time. In contrast to BPQ-BA, the IATS does not demonstrate ambiguity nor bias toward negative appraisals for uncomfortable sensations [79]. Therefore, high IATS scores may provide more appropriate indications of hypervigilance and maladaptive attention toward interoceptive sensations.

Although a maladaptive interoceptive attention style has been described as encompassing hypervigilance and somatisation [11, 12], there is value in distinguishing maladaptive, negatively biased interpretation from attention, which may be separate, albeit related constructs [17, 31]. Relative to BPQ-BA, BPQ-Reactivity is less frequently employed in studies examining self-reported interoception and alexithymia and within broader interoceptive research [26]. However, it merits consideration where researchers seek to operationalise maladaptive interpretation of interoceptive sensations. This proposal is supported by stronger positive correlations for BPQ-R scales with global alexithymia, DIF, and DDF, such that heightened autonomic nervous system reactivity, expressed through perceptions of symptomatology, covary with alexithymia—particularly in impaired identification and expression of feelings. These propensities may perpetuate somatosensory amplification, heightened sympathetic 'fight-or-flight' responses, and stress reactivity [33, 68], potentially rendering some individuals unable to differentiate bodily sensations as distinctly representative of emotion states from symptoms necessitating medical attention.

Whilst limited findings were presented for the SAQ [37], positive relationships were observed with alexithymia. The SAQ assesses negative feelings propensity [26], based on heightened perceptions of experiencing uncomfortable, noxious, and symptomatic bodily sensations, which relates to stronger illness anxiety beliefs. It is tenable that this scale taps into negatively biased evaluations of sensations, given that greater endorsement is reflective of heightened sympathetic 'fight-or-flight' responses [105] which constitutes a key component of anxiety and somatic symptom disorder [17]. Together, BPQ-Reactivity and SAQ may reflect tendencies toward maladaptive, negative categorisation and interpretation of internal signals [31], predisposing individuals to indiscriminately generalise sensations within vague, negatively valenced terms, thereby contributing to greater difficulties with granular identification and articulation of emotions. Such tendencies could propagate interpretations of benign or ambiguous sensations as threatening [17] or indicative of illness, particularly when accompanied by maladaptive interoceptive attention propensities.

Following recent proposals [31] and the multidimensional framework of the original measure [34], we suggest that adaptive interoceptive interpretation in the context of alexithymia involves subdomains assessable by MAIA scales. This domain encompasses positive, non-judgemental appraisals of internal bodily sensations, including emotional interpretations. Considering the present findings and theoretical proposals, adaptive interpretation of interoceptive sensations seems underpinned by interoceptive trusting (MAIA-Trusting) not-worrying (MAIA-NW), emotional awareness (MAIA-EA), and body listening (MAIA-BL). Such aspects constitute interpretational factors, tacitly involving beliefs which may modify perceptions of bodily sensations [31, 34]. Recognising that emotions have a physical component (MAIA-EA) is arguably reflective of an interoceptive belief, whereas actively listening to the body for insight (MAIA-BL) involves attributing meaning to internal bodily signals to inform emotional experiences and decision-making. Moreover, the tendency to not categorise noxious and uncomfortable sensations as worrying or distressing (MAIA-NW) reflects an interpretational style. Additionally, perceptions that enable regarding the body as safe (MAIA-Trusting) reflect positive attitudes regarding internal signals [34]. Collectively, such interpretations may promote the enactment of adaptive behaviours aimed at addressing the perceived state of the body.

## Interoceptive constructs and measurement

The systematic review findings indicate there is consistent use of the term 'interoceptive sensibility', as defined by Garfinkel et al. [28], in studies concurrently examining alexithymia. Use of this term coincided with inconsistent employment of measures to operationalise the construct. This supports observations regarding tendencies for researchers to assign any interoceptive self-report scale to this term [26, 31]. However, frequent use of 'interoceptive awareness' was also noted. Review of how this construct was defined in investigations indicated the persistence of conflicting and competing definitions in the literature. Some studies administered the MAIA, drawing on Mehling et al.'s [34, 35] conceptualisation of the construct. Mehling et al. [35] noted that 'interoceptive awareness', defined by Garfinkel et al. [28], is reductionistic and proposed it should entail "the conscious level of interoception with its multiple dimensions potentially accessible to self-report" (p. 2). Whilst this definition acknowledges the breadth of consciously accessible dimensions, it is descriptive and similarly constitutes an umbrella term. Alternative frameworks have been proposed since this time, which are more precise in dimension delineation and definitions [17, 27, 31]. A parallel emerging trend toward utilising more specific terms (i.e., 'self-reported interoceptive attention and accuracy [27]) was also identified, whereby greater consistency in operationalisation was observed. This hopefully reflects increasing recognition of the importance of construct-measurement convergence in interoceptive research.

## Implications

Recent theoretical proposals have suggested conceptualising interoception hierarchically [17] and to differentiate constructs at different levels of specificity [31]. Considering the present evidence, we advocate for greater precision in specifying the measured interoceptive construct, according to self-report, echoing the need for clearer terminology to enhance convergence between measures and improve generalisability, validity, and replicability of findings. For investigating self-reported maladaptive interoceptive sensing, we recommend the use of the ICQ, ISQ, and EDI-IAw, while the IAS BAQ, MAIA-Noticing, and THISQ are suitable for assessing adaptive interoceptive sensing beliefs. The IATS is proposed for assessing maladaptive attentional tendencies, whereas adaptive attentional processes can be measured through MAIA scales such ND, AR, and SR. For maladaptive interoceptive interpretation, BPQ-Reactivity and SAQ are appropriate, reflecting negatively biased processing of unpleasant sensations. Conversely, adaptive interoceptive interpretation can be assessed using NW, EA, BL, and Trusting MAIA scales. Moreover, it is proposed that certain BAQ items (i.e., from 'Predict Body Reactions', 'Onset of Illness' subscales) may assess adaptive interoceptive memory. However, further validation is needed to confirm whether the recommended self-reports effectively capture these domains, particularly in clinical samples. Future research should test the construct validity of the proposed framework to establish whether adaptive and maladaptive interoception represent second-order constructs influencing sensing, attention, interpretation, and memory, assessed by these interoceptive measures. The meta-analytic findings suggest that the questionnaires capture distinct beliefs of sensing, attention, and interpretation, and should not be uniformly applied to multiple questionnaires. Echoing the advice of Desmedt et al. [31], we recommend adopting a conservative approach, assuming that these measures assess different constructs until convergent evidence is established.

 The included studies predominantly focused on global alexithymia; however, the multifaceted nature of alexithymia was evident in the present findings. Interoceptive self-reports showed stronger associations with DIF and DDF compared to EOT. Researchers are therefore encouraged to consider alexithymic facets as separate outcomes when investigating these constructs.

This approach could provide further insights into which interoceptive schemas are associated with difficulties appraising emotions and whether self-reported interoceptive sensing, attention, interpretation, and memory impact on externally directed attention styles.

Issues concerning construct measurement were also identified during the review, particularly concerning computation of MAIA-Total scores [56, 60, 61, 64, 73, 75, 76, 79]—variably conceptualised as representing global interoceptive sensibility [32] or adaptive interoception [26]. Mehling and colleagues [34] observed poor fit for a single-factor model and discouraged calculation of summary scores. However, recent evidence indicates the existence of a general MAIA factor, supporting computation of a summary score to index global adaptive interoception. This score should consist of Noticing, AR, EA, SR, BL, and Trusting scales [26, 32, 38]. Despite this, no recent studies computing MAIA-Total reported omission of ND and NW scales from calculation. This is problematic, as these scales have demonstrated heterogeneity relative to the other six [26, 32, 38]. It is recommended to exclude ND and NW from computation when analysing MAIA-Total scores to ensure clearer interpretation of overall effects.

Although the current findings primarily concerned non-clinical samples, they are of clinical relevance, considering that interoceptive and emotional dysfunctions underlie various disorders [106]. There is emerging consideration of interoception in targeted mind-body therapeutic interventions that enable reframing interoceptive interpretations and regulation of autonomic reactivity [107]. However, alexithymic traits, such as DIF and DDF, can significantly influence treatment outcomes following intervention for psychiatric disorders [46]. Overall, our meta-analyses indicated that these alexithymic traits coincide with perceived dysfunctional, maladaptive interoceptive schemas—findings supported by evidence for convergence of these constructs [67]. Accordingly, assessing interoception and alexithymia through self-report scales seems beneficial. This approach would inform holistic case conceptualisations, providing insights into an individual's interoceptive and emotional awareness. Measuring interoception and alexithymia may facilitate the delivery of therapies aimed at strengthening adaptive interoceptive beliefs, which could enhance capacities for identifying and articulating emotions. In turn, improvement of interoceptive beliefs and emotional skills could bolster adaptation to dynamic environmental stressors and challenges.

## Limitations

Firstly, we recognise the susceptibility of self-report scales to biased responses. However, self-reports remain invaluable for assessing subjective experiences, as people vastly vary in their perception and communication of physiological and emotional feelings [95, 108]. Therefore, we deemed it meaningful to review relationships between various self-reports and alexithymia. This is supported by recent calls for clarification of key differences in interoceptive self-report scales [12] which provides useful indication of clinical status when compared to behavioural indicators [18]. The proposed framework serves as a singular suggestion for enhancing construct-measurement convergence. This may provide a foundation for future investigations regarding alternative classification methods concerning the constructs measured by interoceptive self-reports, such as the valence of included sensations. The aim of such endeavours, however, remains consistent: reducing discrepancies in interoceptive domain definitions and their corresponding measurements. We also acknowledge that aspects of inter-rater reliability (see S2 File) were suboptimal—particularly for the title and abstract screening phase. As such, replicability of this phase could be problematic. Moreover, for the extraction phase, whilst percentage agreement suggested a high level of reliability, our reliance on percentage alone could overestimate true reliability. Various measures were noted as administered more frequently (e.g., TAS-20 cf. PAQ; MAIA cf. BAQ). Disparities in the studies included in meta-analyses

may have influenced our results. Therefore, some caution should be exercised when interpreting these findings. Moreover, high heterogeneity was observed in several analyses, particularly with the ICQ, which lacks robust psychometric properties and formal validation [41]. This complicated interpretation of the meta-analytic findings, subsequently affecting confidence in the pooled estimated correlations. Despite efforts to conduct pre-registered subgroup analyses according to alexithymic scale, clinical status, and regional location of samples, many analyses were underpowered. Although interesting and significant differences according to geographic variability were identified in our secondary meta-analyses, overall, the smaller sample sizes in certain subgroups (PAQ, BVAQ, clinical, Australasia, North America, and Asia) potentially limited our ability to detect significant subgroup differences. Consequently, conclusively determining whether these factors impact the relationship between self-reported interoception and alexithymia was hindered. Clarifying these influences therefore remains an important pursuit. Lastly, we acknowledge factors affecting inclusion of potentially eligible studies and effects. We did not contact authors of eligible articles for unreported effects, due to previous poor response rates [49, 50]. Moreover, although 'interoceptive awareness' was commonly investigated across articles, this was identified during data extraction. Ideally, this construct would have been included in search terms across databases. As such, eligible articles were likely not identified, and relevant effects excluded.

## Conclusion

This study systematically reviewed and meta-analysed the relationship between specific interoceptive self-report scales and distinct aspects of alexithymia. Findings revealed inconsistencies in the conceptualisation and operationalisation of subjective interoceptive constructs across studies using self-report measures of interoception and alexithymia, suggesting a need for more precise terminology. These observations are corroborated by meta-analyses demonstrating that the relationship between self-reported interoception and alexithymia differs as a function of the interoceptive construct measured by self-report. We found that questionnaires proposed to assess maladaptive forms of interoceptive sensing (ICQ, ISQ, EDI-IAw), attention (IATS), and interpretation (BPQ-Reactivity, SAQ) were positively associated with alexithymia, while scales purportedly measuring adaptive aspects of sensing (IAS, BAQ, MAIA-Noticing, THISQ), attention (MAIA-ND, MAIA-AR, MAIA-SR), interpretation (MAIA-NW, MAIA-EA, MAIA-BL, MAIA-Trusting), and memory (BAQ) had overall negative associations. As such, these interoceptive constructs either reinforce or reduce alexithymia—namely DIF and DDF facets. This study highlights that specificity and precision in labelling and measuring interoceptive constructs is an essential first step towards addressing discrepancies in interoceptive construct definitions and accompanying measurements. Researchers and clinicians are therefore encouraged to employ suitable questionnaires to measure specific interoceptive constructs. Self-reported interoceptive deficits are a highly relevant feature of alexithymia and deserve consideration as a contributing factor to mental and physical disorders. Accordingly, assessment of self-reported interoception and alexithymia at global and facet levels is suggested. Therapeutically targeting interoceptive mechanisms may improve emotional awareness and articulation capacities, enhance mind-body connections, and improve treatment outcomes for patients.

## Supporting information

**S1 File. PRISMA 2020 checklist.**
(DOCX)

**S2 File. Inter-rater reliability.**
(DOCX)

**S3 File. Risk of bias assessment.**
(DOCX)

**S4 File. Characteristics of included studies.**
(DOCX)

**S5 File. Interoceptive self-report scales employed in included studies.**
(DOCX)

**S6 File. Sample characteristics and extracted correlations of each independent sample within included studies employing interoceptive self-report scales to examine their relationship with global alexithymia.**
(DOCX)

**S7 File. Sample characteristics and extracted correlations of each independent sample within included studies employing interoceptive self-report scales to examine their relationship with DIF.**
(DOCX)

**S8 File. Sample characteristics and extracted correlations of each independent sample within included studies employing interoceptive self-report scales to examine their relationship with DDF.**
(DOCX)

**S9 File. Sample characteristics and extracted correlations of each independent sample within included studies employing interoceptive self-report scales to examine their relationship with EOT.**
(DOCX)

**S10 File. Covidence export of screened articles.**
(PDF)

## Author Contributions

**Conceptualization:** Kristen Van Bael, Jessica Scarfo, Emra Suleyman, Michelle Ball.

**Data curation:** Kristen Van Bael, Jessica Katherveloo, Natasha Grimble.

**Formal analysis:** Kristen Van Bael.

**Investigation:** Kristen Van Bael, Jessica Katherveloo, Natasha Grimble.

**Methodology:** Kristen Van Bael.

**Project administration:** Kristen Van Bael.

**Supervision:** Jessica Scarfo, Emra Suleyman, Michelle Ball.

**Validation:** Kristen Van Bael.

**Visualization:** Kristen Van Bael.

**Writing – original draft:** Kristen Van Bael.

**Writing – review & editing:** Kristen Van Bael, Jessica Scarfo, Emra Suleyman, Michelle Ball.

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
