## [Decision Letter · Decision Letter 0]

3 Jul 2024

PONE-D-24-17726A systematic review and meta-analysis of the relationship between subjective interoception and alexithymia: Implications for construct definitions and measurementPLOS ONE

Dear Dr. Van Bael,

Thank you for submitting your manuscript to PLOS ONE. After careful consideration, we feel that it has merit but does not fully meet PLOS ONE’s publication criteria as it currently stands. Therefore, we invite you to submit a revised version of the manuscript that addresses the points raised during the review process.

Overall, the feedback from all three reviewers was extremely positive. This manuscript is an important, comprehensive, and methodologically sound endeavor to provide in-depth insights into the association between alexithymia and interoception.

From a methodological standpoint, there are no major issues to resolve, except for providing inter-rater reliability for study selection and data extraction, as recommended by several systematic review and meta-analysis guidelines.

From a conceptual standpoint, there are some important topics that need to be clarified or discussed in the manuscript, particularly regarding the criteria for classifying questionnaires and how alternative classification approaches could also be feasible.

Lastly, despite the manuscript being extremely well-written, I believe there is still some room for improvement, particularly considering the complexity of the analysis and results. The reviewers provide several comments that can be useful for the authors to make the manuscript clearer to the reader (e.g., summary figure).

A rebuttal letter that responds to each point raised by the academic editor and reviewer(s). You should upload this letter as a separate file labeled 'Response to Reviewers'.A marked-up copy of your manuscript that highlights changes made to the original version. You should upload this as a separate file labeled 'Revised Manuscript with Track Changes'.An unmarked version of your revised paper without tracked changes. You should upload this as a separate file labeled 'Manuscript'.We look forward to receiving your revised manuscript.

Kind regards,

Carlos Campos

Academic Editor

PLOS ONE

Journal Requirements:

2. We note that your Data Availability Statement is currently as follows: [All relevant data are within the manuscript and its Supporting Information files]

Reviewers' comments:

Reviewer's Responses to Questions

**Comments to the Author**

1. Is the manuscript technically sound, and do the data support the conclusions?

Reviewer #1: Yes

Reviewer #2: Yes

Reviewer #3: Yes

2. Has the statistical analysis been performed appropriately and rigorously? 

Reviewer #1: Yes

Reviewer #2: N/A

Reviewer #3: Yes

3. Have the authors made all data underlying the findings in their manuscript fully available?

Reviewer #1: Yes

Reviewer #2: Yes

Reviewer #3: Yes

4. Is the manuscript presented in an intelligible fashion and written in standard English?

Reviewer #1: Yes

Reviewer #2: Yes

Reviewer #3: Yes

5. Review Comments to the Author

Reviewer #1: This systematic review and these meta-analyses are excellent for several reasons. The authors greatly explained the actual state of the literature on self-report interoception and followed the most up-to-date recommendations in the field. The method is overall very good. The discussion is complete and insightful for the field. Once published, this review will have a huge and, more importantly, a very positive impact on the literature, as it promotes valid and nuanced interpretations of data. I have rarely read such a well-written manuscript. Many congrats to the authors!

You can find below my minor comments:

1. Please report the inter-rater reliability for the selection of papers, their extraction, and their assessment.

2. There seems to be a typo in this sentence as the correlation between ICQ and alexithymia is reported twice with different results: “Meta-analysis demonstrated a strong positive association for the ICQ and alexithymia, r(9) = 0.57, p <.001, suggesting greater struggles to interpret non-affective interoceptive states coincides with difficulties identifying and describing feelings, and a preference for externally oriented thinking. A strong positive effect was further observed between the ICQ and alexithymia, r(2) = 0.53, p <.001, indicating that greater difficulty registering or interpreting interoceptive sensations relates to higher alexithymia.”

3. I do not agree that the THISQ is a measure of neutral interoceptive attention, as attentional aspects are not mentioned in the items of this questionnaire. Neutral interoceptive detection would be more relevant. Similarly, I don’t think that the BAQ is a measure of adaptive interoceptive attention. Items are more related to the capacity to notice and predict body reactions to internal and external factors such as weather, seasons, foods, blows, diseases, and energy level. In sum, the THISQ and the BAQ are questionnaires assessing the capacity to detect, notice, or predict bodily cues that should be detected/predicted for maintaining well-being or homeostasis (i.e., adaptive interoceptive abilities).

4. The authors propose to differentiate questionnaires based on their adaptivity aspects. Three comments: (1) Could the authors further discuss their definition of this criterion? (2) Could we categorize these questionnaires differently (e.g., based on the valence of sensations reported in the questionnaire)? (3) They should recommend future studies to test the validity of this classification into two categories (i.e., maladaptive vs. adaptive).

Reviewer #2: I commend the authors for what was clearly a massive undertaking. This area, as they clearly articulate, is a like a field of land-mines and they navigated it gracefully. I think, once cleaned up, this work will represent a clear contribution to research on interoception and alexithymia, as well as to research on emotional expertise more broadly. Really it is a tour-de-force. Please interpret my comments as constructive! I would ultimately love to see this published.

Overall comments:

1. I'll note first and foremost that I do not have personal experience conducting meta-analyses so I will leave it to another reviewer to comment on whether statistical analysis have been performed appropriately and rigorously. However I do want to applaud the careful approach the authors took to data disaggregation. I especially appreciated that the authors attempted to examine variability in effects across geographic regions.

2. I do want to acknowledge with full transparency that I had some difficulty following the logic of several sections. I recognize that it must be very difficult organizing so many theoretical perspectives, constructs, and ultimately, findings. I cant help but wonder if a bit more structure (e.g., headings) or sign-posting (e.g., in each results section we break down results based on effect size) would help the reader keep things straight in their mind. I also found several broken or grammatically messy sentences throughout, so I would encourage the authors to do another round of careful editing.

Additional comments:

3. I think the authors need to be a bit more explicit about how their paper contributes above and beyond what's already been published by Trevisan et al. To be clear, I think this paper DOES contribute above and beyond what's already been published, but I think it wouldn't hurt to make this more explicit RIGHT AWAY (before overviewing Trevisan et al's specific findings). When you introduced the paper (which I hadn't read), my first thought was, "wait, did someone do this already?"

4. Related to point 2 above, in the introduction, the authors discuss (a) popular frameworks for organizing interoceptive CONSTRUCTS and (b) the various MEASURES used in interoceptive research with the goal (as I understand it) being to clarify how these two things are somewhat incoherent and poorly aligned. I wonder if using the language of "constructs" versus "measures" (i.e., a more explicit construct validity framework) would be helpful. As it is, the introduction mixes up discussions of both (a) and (b) in a way that was difficult to follow.

5. I recognize the need for acronyms but I would encourage the authors to go through and reduce their use of acronyms wherever possible. Particularly in the discussion, I found the prevalence of acronyms made it difficult to follow the narrate arc of each section. Acronyms should also be spelled out in footnotes for tables and figures. Especially given the complexity of your findings. Someone should be able to look at your figures and get the whole story.

6. Please add a mean and standard deviation for participant ages if possible. 18-91 is a very broad age range and, given evidence that both self-report and task-based measures of interoception vary across the lifespan, this seems like a relevant detail.

7. On line 331, can you clarify what you mean by "self-reported psychiatric diagnoses"? Are you distinguishing between formal and informal diagnosis here?

8. Take this or leave it -- I personally think risk of bias assessments can go to the supplemental so that the main analyses about relationships among measures has a bit more breathing room.

9. Line 527, you say "meta-analysis demonstrated a strong positive association for the ICQ and alexithymia..." and then on 530, "A strong positive effect was further observed between the ICQ and alexithymia..." I assume this is a mistake?

10. Throughout the paper, and in the discussion in particular, be careful when you use "interoceptive accuracy" since that will conjure up images of behavioral tasks for many readers given Garfinkel et als work. I would be very clear when you're referring to "task-based interoceptive accuracy" (e.g., on line 777?) vs self-reported interoceptive accuracy.

11. Line 835, you say, "DIF exhibited the highest magnitudes in associations with interoceptive self-report scales, which could be due scale construction and emotion generation processes. " I know this sentence has a typo, but on top of that I found it a bit confusing. Maybe consider something like, "which could be due to specific aspects of scale construction or to more general features of emotion generation processes. For example, ..."

13. On line 841 you write, "Additionally, theories of emotion propose that the interpretation of physiological sensations and changes plays an essential role in the generation of emotions, as they may initiate an affective state." The constructionist model (whose proponents you cite) would argue that "physiological sensations and changes" partially CONSTITUTE core affect. They would not argue that these sensations INITIATE affect. This is an important detail that these folks may be finicky about.

14. I'm not sure i'm clear on your logic why the THISQ is a measure of "neutral interoceptive attention." You never really describe what you mean by "neutral."

15. Would love to see a sentence or two on geographic variability given that you ran those analyses. Otherwise I would omit them from reporting and just say somewhere higher up in the results or methods that you ran those pre-registered subgroup analyses but have reported them in the supplement given small sample sizes. In general, I think any analyses that don't contribute to the main analyses (which are already complex) should be placed in the supplement.

Reviewer #3: Van Bael et al., conducted a thorough systematic review and several meta-analyses of currently used self-report interoceptive and alexithymia measures. Strengths of this research include the pre-registration and clear approach to assess the different measures used within the literature. I think that this study adds value to the current literature as it adds clarity to a field which is limited by its lack of consistency in definitions and measures. Further, the study is well-written. Therefore, my comments are minor, and I think that this will be a useful work to guide future researchers interested in investigating interoception and alexithymia.

• In the eligibility criteria, I think that it should be noted that clinical disorders were considered (i.e., they were not excluded).

• Given that the meta-analyses produces a lot of results, a summary figure/s would also enhance this study and help readers to understand the overall picture. This could be for example in the form of a spider plot or heat map showing the negative and positive correlations between each alexithymia domain and the interoceptive measures.

• Revise manuscript for repetition in results to increase conciseness. For example,

-Regarding the primary meta-analyses: can authors provide a workflow or diagram which summarises how many meta-analyses were conducted across the alexithymia scales? Instead of repeating this at the beginning of each subheading of the results (i.e., line 521, 582, 671).

-Lines: 568-571; 620-622; 695-697: regarding subgroup analyses. This sentence is repeated throughout the results but should be placed in the statistical analysis section of the methods and stated once.

- Broadly, I think that references to the statistical thresholds and effect size cut-offs could be moved to the methods (Line 510-518), together with the information about the sub-group analyses.

• In discussion, avoid using “positive” or negative” association and instead explain the relationship “i.e., greater alexithymia was associated with higher levels of interoceptive confusion” Line 754.

• Some measures are more popular or well-known than others (i.e., TAS-20, MAIA) and therefore may have had more weight in each meta-analysis. Authors should mention how this might have influenced their results, as some frequencies of use = 1 (i.e., Figure 2), so I would think that these results need to be interpreted with some caution.

Minor comments:

Line 174: wording “as meta-analysis” change for “meta-analytic approaches should enable the quantification”

Line 245: wording “meta-analysed”

Line 281: Typo “Supplemental”

Line 380: Move the section of excluded scales to end of section to improve flow.

Line 714: Avoid using the word “forced” due to negative connotations. Replace with more neutral word like “used” or “employed”

Line 1076-1078: Link results of current study to this sentence as it is currently not clear that using appropriate measures of interoception and alexithymia will likely improve therapeutic outcomes.

Figure 1: check figure quality

6. PLOS authors have the option to publish the peer review history of their article (what does this mean?). If published, this will include your full peer review and any attached files.

Reviewer #1: No

Reviewer #2: No

Reviewer #3: **Yes: **Jessica L. Hazelton

---

## [Author Response · Author response to Decision Letter 0]

25 Jul 2024

Response to Editor and Reviewers

Editor’s Comments

From a methodological standpoint, there are no major issues to resolve, except for providing inter-rater reliability for study selection and data extraction, as recommended by several systematic review and meta-analysis guidelines.

 We confirm that inter-rater reliability for study selection, data extraction, and risk of bias have been provided. Due to the length of the manuscript, we chose to summarise these results in the main reporting, with full inter-rater reliability results reported in the revised Supporting Information (File S2).

From a conceptual standpoint, there are some important topics that need to be clarified or discussed in the manuscript, particularly regarding the criteria for classifying questionnaires and how alternative classification approaches could also be feasible.

 We have provided clearer definitions for adaptive and maladaptive interoception in the Introduction (p. 2, L 62-73), which we feel better clarifies our reasoning for classifying questionnaires in this way. Whilst we believe that the suggestion to classify questionnaires according to the valence of assessed sensations is valuable, we feel that our classification provides a nuanced account for the reviewed measures, supported by existing and emerging interoceptive frameworks and their emotional outcomes following our meta-analytic findings. However, we have specified that our findings may provide a foundation for alternative, appropriate classification systems in future, such as the valence of sensations in self-reports.

Lastly, despite the manuscript being extremely well-written, I believe there is still some room for improvement, particularly considering the complexity of the analysis and results. The reviewers provide several comments that can be useful for the authors to make the manuscript clearer to the reader (e.g., summary figure).

 We strongly agreed with this suggestion from the Reviewers. We have added a workflow diagram (Fig 4) and a summary figure of the meta-analysis correlations (heat map, Fig 5).

Reviewers' comments

Reviewer #1

This systematic review and these meta-analyses are excellent for several reasons. The authors greatly explained the actual state of the literature on self-report interoception and followed the most up-to-date recommendations in the field. The method is overall very good. The discussion is complete and insightful for the field. Once published, this review will have a huge and, more importantly, a very positive impact on the literature, as it promotes valid and nuanced interpretations of data. I have rarely read such a well-written manuscript. Many congrats to the authors!

Many thanks to the reviewer for their kind and considered feedback on the paper, especially in terms of its impact on the field! It is greatly appreciated. We have provided our responses to each of their comments and suggestions below.

You can find below my minor comments:

1. Please report the inter-rater reliability for the selection of papers, their extraction, and their assessment.

We concur that this is necessary information to report and apologise for the omission. Due to the length of this paper, we have provided a summary of agreements/reliability amongst reviewers in the paper (Results - Study Selection; Methods – Risk of bias assessment), with more detailed information regarding inter-rater reliability for study selection, data extraction, and risk of bias as a new Supporting Material (File S2).

2. There seems to be a typo in this sentence as the correlation between ICQ and alexithymia is reported twice with different results: “Meta-analysis demonstrated a strong positive association for the ICQ and alexithymia, r(9) = 0.57, p <.001, suggesting greater struggles to interpret non-affective interoceptive states coincides with difficulties identifying and describing feelings, and a preference for externally oriented thinking. A strong positive effect was further observed between the ICQ and alexithymia, r(2) = 0.53, p <.001, indicating that greater difficulty registering or interpreting interoceptive sensations relates to higher alexithymia.”

Many thanks to the reviewer for identifying this typo! This has been amended to report results for the ICQ and ISQ in terms of their association with global alexithymia. (p. 24, L 653).

3. I do not agree that the THISQ is a measure of neutral interoceptive attention, as attentional aspects are not mentioned in the items of this questionnaire. Neutral interoceptive detection would be more relevant. Similarly, I don’t think that the BAQ is a measure of adaptive interoceptive attention. Items are more related to the capacity to notice and predict body reactions to internal and external factors such as weather, seasons, foods, blows, diseases, and energy level. In sum, the THISQ and the BAQ are questionnaires assessing the capacity to detect, notice, or predict bodily cues that should be detected/predicted for maintaining well-being or homeostasis (i.e., adaptive interoceptive abilities).

We thank the reviewer for their insightful comments regarding the THISQ and BAQ and appreciate the opportunity to clarify our discussion of these measures. We acknowledge your point that the THISQ possibly assesses interoceptive detection. Indeed, Vlemincx and colleagues (2023, https://www.tandfonline.com/doi/full/10.1080/08870446.2021.2009479) characterise the THISQ as a measure of self-reported perception of neutral respiratory, cardiac, and gastroesophageal sensations. In their discussion of evidence for convergent validity, Vlemincx et al. further note that THISQ scores correlated moderately with other measures of non-emotional interoceptive attention, such as the BAQ and the Noticing scale of the MAIA. Considering the congruence between the authors’ and our interpretations, we feel that ‘neutral attention’ best characterises the THISQ, in accordance with current evidence. This information has been added to the Discussion (p. 44, L 1185). We have also provided further reasoning for what the authors of THISQ mean by neutral in its subsection within Administered Measures (p. 21, L 541-543), and hope this clarifies the logic behind classifying the questionnaire a measure of neutral attention. 

The BAQ is currently classified as a self-report of interoceptive attention in the literature (e.g., Trevisan et al. 2020, https://onlinelibrary.wiley.com/doi/10.1002/aur.2458; Vlemincx et al. 2023), as noticing and frequency of awareness involves orienting attention to salient interoceptive signals (Mehling, 2012, https://journals.plos.org/plosone/article?id=10.1371/journal.pone.0048230), therein a key subfactor of interoceptive attention (Desmedt et al., 2023). Due to these factors, we are inclined to comply with current opinion and evidence supporting the BAQ’s classification as a measure of interoceptive attention. However, we concur with your view that beliefs (e.g., prediction) assessed in the BAQ involve the integration of external and internal information. Whilst we did not have data examining BAQ subscales (e.g., ‘predict body reactions’) and alexithymia, we have adjusted our discussion to highlight the BAQ's potential role in assessing both adaptive interoceptive accuracy and attention beliefs. (p. 44, L 1175-1182).

4. The authors propose to differentiate questionnaires based on their adaptivity aspects. Three comments: (1) Could the authors further discuss their definition of this criterion? (2) Could we categorize these questionnaires differently (e.g., based on the valence of sensations reported in the questionnaire)? (3) They should recommend future studies to test the validity of this classification into two categories (i.e., maladaptive vs. adaptive).

Our responses to each of these great comments are as follows:

1. We have provided clarification on definitions of adaptive and maladaptive interoception in the Introduction (pp. 4-5, L 63-74), which contextualises later differentiation of adaptive and maladaptive forms of attention, accuracy, and interpretation, according to their impact on efficient physiological regulation (e.g., homeostasis/allostasis).

2. The categorisation of questionnaires based on the valence of sensations assessed is a thoughtful and considered suggestion from the Reviewer. While we understand the rationale behind this suggestion, we believe that categorising the questionnaires in this manner may not fully capture more specific constructs. For instance, several subscales from the MAIA contain items involving pleasant, unpleasant, and neutral sensations. It would be challenging to categorise these subscales according to sensation valence. However, the valence of sensations (e.g., pleasant, neutral, unpleasant) was considered in our differentiation of adaptive versus maladaptive interoceptive abilities, focusing on accuracy, attention, and interpretation. The differentiation of adaptive and maladaptive questionnaires is rooted in the emotional and regulatory outcomes associated with the interoceptive abilities they measure. While we acknowledge that the valence of sensations tacitly informed our differentiation (e.g., for measures such as the SAQ and BPQ Reactivity scales), we believe that the current categorisation provides a nuanced framework for understanding these measures, supported by emergent frameworks developed to improve the conceptualisation and measurement of interoception (e.g., Desmedt et al., 2023). We hope this explanation clarifies our position but acknowledge that the suggestion is a valuable one. We have adjusted our Discussion to clarify that our classification is merely one way to address the current inconsistencies in conceptualisation and measurement, suggesting that classification based on valence of sensations included in questionnaires may be another avenue worth pursuing (pp. 49-50, L 1329-1354).

3. We completely agree and have expanded on this in the Implications section (p. 48, L 1281-1284).

Reviewer #2

I commend the authors for what was clearly a massive undertaking. This area, as they clearly articulate, is a like a field of land-mines and they navigated it gracefully. I think, once cleaned up, this work will represent a clear contribution to research on interoception and alexithymia, as well as to research on emotional expertise more broadly. Really it is a tour-de-force. Please interpret my comments as constructive! I would ultimately love to see this published.

We thank the Reviewer for their positive feedback on our endeavour, and greatly appreciate their considered feedback on aspects to improve the quality of our paper. We have provided our responses to each of their comments and suggestions below.

Overall comments:

1. I'll note first and foremost that I do not have personal experience conducting meta-analyses so I will leave it to another reviewer to comment on whether statistical analysis have been performed appropriately and rigorously. However I do want to applaud the careful approach the authors took to data disaggregation. I especially appreciated that the authors attempted to examine variability in effects across geographic regions.

Many thanks to the reviewer for their comments regarding our methodological approaches!

2. I do want to acknowledge with full transparency that I had some difficulty following the logic of several sections. I recognize that it must be very difficult organizing so many theoretical perspectives, constructs, and ultimately, findings. I cant help but wonder if a bit more structure (e.g., headings) or sign-posting (e.g., in each results section we break down results based on effect size) would help the reader keep things straight in their mind. I also found several broken or grammatically messy sentences throughout, so I would encourage the authors to do another round of careful editing.

We thank the author for their transparency in their experience reading the paper. We have done another round of editing and feel that the paper reads clearer. We also felt that the current headings are sufficient for containing results pertinent to each alexithymic outcome. We have therefore provided clearer signposting in the introductory paragraph of the ‘The relationship between self-reported interoception and alexithymia’ results section, per the Reviewer’s suggestion (p. 23, L 602-603). We believe that this will enhance readability and keeping track of the results.

Additional comments:

3. I think the authors need to be a bit more explicit about how their paper contributes above and beyond what's already been published by Trevisan et al. To be clear, I think this paper DOES contribute above and beyond what's already been published, but I think it wouldn't hurt to make this more explicit RIGHT AWAY (before overviewing Trevisan et al's specific findings). When you introduced the paper (which I hadn't read), my first thought was, "wait, did someone do this already?"

We greatly appreciate this feedback. We have added a paragraph to describe how this study contributes above and beyond already published findings in the Introduction (p. 4, L 83-92). 

4. Related to point 2 above, in the introduction, the authors discuss (a) popular frameworks for organizing interoceptive CONSTRUCTS and (b) the various MEASURES used in interoceptive research with the goal (as I understand it) being to clarify how these two things are somewhat incoherent and poorly aligned. I wonder if using the language of "constructs" versus "measures" (i.e., a more explicit construct validity framework) would be helpful. As it is, the introduction mixes up discussions of both (a) and (b) in a way that was difficult to follow.

We thank the Reviewer for this feedback and have updated the Introduction and Discussion sections to be more explicit in the discussion of constructs versus measures across the various frameworks. We have maintained some terminology (e.g., ‘factors’ and ‘subfactors’) according to the authors’ proposals per their frameworks. 

5. I recognize the need for acronyms but I would encourage the authors to go through and reduce their use of acronyms wherever possible. Particularly in the discussion, I found the prevalence of acronyms made it difficult to follow the narrate arc of each section. Acronyms should also be spelled out in footnotes for tables and figures. Especially given the complexity of your findings. Someone should be able to look at your figures and get the whole story.

We thank the Reviewer for their openness in describing their experience reading the paper. Whilst many of the questionnaire acronyms are commonly employed in interoceptive research, you have highlighted something that also made us uncomfortable. We agree that the use of acronyms is extensive; however, the paper is already very long. Using the full names of the questionnaires would undoubtedly add to its length. Accordingly, we have decided to maintain the acronyms. We have ensured that an acronyms list and questionnaire descriptions has been provided as a Supporting Material (File S5, Table S4), and that all acronyms have been included in both the table footnotes and figures throughout the revised paper. Reviewer 3 suggested that the Discussion explains meta-analysis results rather than in terms of ‘positive’ and ‘negative’ relationships. The previous structuring of the discussion in this way was laden with acronyms. We have adjusted the Discussion to provide context for the questionnaire acronyms. Together, we feel that these compromises and changes will ensure the narrative arc can be followed.

6. Please add a mean and standard deviation for participant ages if possible. 18-91 is a very broad age range and, given evidence that both self-report and task-based measures of interoception vary across the lifespan, this seems like a relevant detail.

We thank the reviewer for their suggestion. We note that mean ages for samples have been provided in File S4 of the Supporting Information with the resubmission, which provides expanded characteristics of the included studies.

7. On line 331, can you clarify what you mean by "self-reported psychiatric diagnoses"? Are you distinguishing between formal and informal diagnosis here?

We have clarified that the sample from this paper (Murphy et al.

---

## [Decision Letter · Decision Letter 1]

13 Aug 2024

PONE-D-24-17726R1A systematic review and meta-analysis of the relationship between subjective interoception and alexithymia: Implications for construct definitions and measurementPLOS ONE

Dear Dr. Van Bael,

Thank you for submitting your manuscript to PLOS ONE. After careful consideration, we feel that it has merit but does not fully meet PLOS ONE’s publication criteria as it currently stands. Therefore, we invite you to submit a revised version of the manuscript that addresses the points raised during the review process. I believe that you addressed most of the major concerns expressed by the reviewers in the original revision, but one important conceptual issue is not completely addressed. One of the reviewers questioned the classification of interoception attention measures (particularly THISQ and BAQ) and how this interferes with the conclusions regarding the association between this construct and alexithymia. The comments are sound, well-grounded and extremely important considering the conceptual debate regarding interoception measurement. The authors should consider these comments and either adjust the manuscript accordingly or provide a strong rebuttal addressing the presented concerns. Regardless, the authors should also provide a more in-depth discussion about this issue on the Discussion section so that the lack of a consensual classification is clear for the reader. Lastly, some minor recommendations were given regarding inter-rater reliability, but I believe that these will be easy to acommodate. Please submit your revised manuscript by Sep 27 2024 11:59PM. If you will need more time than this to complete your revisions, please reply to this message or contact the journal office at plosone@plos.org. Please include the following items when submitting your revised manuscript:A rebuttal letter that responds to each point raised by the academic editor and reviewer(s). You should upload this letter as a separate file labeled 'Response to Reviewers'.A marked-up copy of your manuscript that highlights changes made to the original version. You should upload this as a separate file labeled 'Revised Manuscript with Track Changes'.An unmarked version of your revised paper without tracked changes. You should upload this as a separate file labeled 'Manuscript'.If applicable, we recommend that you deposit your laboratory protocols in protocols.io to enhance the reproducibility of your results. Protocols.io assigns your protocol its own identifier (DOI) so that it can be cited independently in the future. For instructions see: https://journals.plos.org/plosone/s/submission-guidelines#loc-laboratory-protocols. Additionally, PLOS ONE offers an option for publishing peer-reviewed Lab Protocol articles, which describe protocols hosted on protocols.io. Read more information on sharing protocols at https://plos.org/protocols?utm_medium=editorial-email&utm_source=authorletters&utm_campaign=protocols.

We look forward to receiving your revised manuscript.

Kind regards,

Carlos Miguel Martins Campos

Academic Editor

PLOS ONE

Journal Requirements:

Reviewers' comments:

Reviewer's Responses to Questions

**Comments to the Author**

1. If the authors have adequately addressed your comments raised in a previous round of review and you feel that this manuscript is now acceptable for publication, you may indicate that here to bypass the “Comments to the Author” section, enter your conflict of interest statement in the “Confidential to Editor” section, and submit your "Accept" recommendation.

Reviewer #1: (No Response)

Reviewer #2: All comments have been addressed

Reviewer #3: All comments have been addressed

2. Is the manuscript technically sound, and do the data support the conclusions?

Reviewer #1: Yes

Reviewer #2: Yes

Reviewer #3: Yes

3. Has the statistical analysis been performed appropriately and rigorously? 

Reviewer #1: Yes

Reviewer #2: Yes

Reviewer #3: Yes

4. Have the authors made all data underlying the findings in their manuscript fully available?

Reviewer #1: Yes

Reviewer #2: Yes

Reviewer #3: Yes

5. Is the manuscript presented in an intelligible fashion and written in standard English?

Reviewer #1: Yes

Reviewer #2: Yes

Reviewer #3: Yes

6. Review Comments to the Author

Reviewer #1: I would like to thank the authors for their exhaustive answers. I, nevertheless, have some remaining comments.

1. The methodology employed to assess inter-rater reliability is overall good, but also reveals some areas for potential improvement. For the title and abstract screening phase, the reported kappa values (κ = 0.10-0.39) indicate slight to fair agreement following the most liberal guidelines (Landis & Koch, 1977), but poor (Fleiss, 1981; Altman, 1991) following more conservative ones. This means that the replicability of this selection phase is not optimal and should be acknowledged in the limitations. For the extraction phase, the percentage agreement shows a high level of reliability (84.4 to 90.6%), which is commendable, but the reliance on percentage agreement alone, without considering chance agreement, may overestimate true reliability. This should also be acknowledged in the limitations.

2. Although I have explained that the THISQ and the BAQ are not measures of interoceptive attention, the authors stand by their interpretation based on the following arguments: (1) The authors of the THISQ (Vlemincx et al., 2023) note that THISQ scores correlated moderately with other measures of non-emotional interoceptive attention, such as the BAQ and the Noticing scale of the MAIA; (2) The BAQ is currently classified as a self-report of interoceptive attention in the literature (e.g., Trevisan et al. 2020; Vlemincx et al. 2023), as noticing and frequency of awareness involves orienting attention to salient interoceptive signals (Mehling, 2012). Here are my answers: (1) the authors of the THISQ wrote: “Self-reported interoception as assessed by the THISQ is defined as the self-reported awareness or observation of these sensations, assessed by the rate at which persons notice or feel (changes in) these internal bodily sensations. By assessing the ‘noticing’ and ‘feeling’ of sensations, we excluded interoceptive features such as discrimination, accuracy and attention, to avoid specific emotional or cognitive interpretations of sensations (similar to the BAQ and BPQ)” (p.1238). They also wrote: “… the THISQ assesses self-reported interoception... However, it does not explicitly assess interoceptive accuracy, nor interoceptive attention… The THISQ items are not subjective evaluations of whether one pays attention to these sensations, nor of whether one perceives these sensations accurately.” (p.1249) A moderate correlation does not support the conclusion that two measures evaluate the same construct. It is expected that a measure of attention correlates with a measure of awareness, as more attention should lead to more awareness; the two constructs are related but distinct. Put differently, while attention is necessary for individuals to notice and be aware of their sensations, it is not sufficient. (2) For the BAQ, it is more debatable as the original authors have themselves qualified this questionnaire as a measure of interoceptive attention. Again, however, although I agree that attention would help to better notice or predict bodily sensations, I do not agree that the BAQ is a direct measure of interoceptive attention given that no item refers to attention per se.

3. This debate about the construct (i.e., whether or not it is interoceptive attention) measured by certain questionnaires is not without consequences. Indeed, the authors conclude that alexithymia is associated with more interoceptive attention. I do not agree. Questionnaires relevant to interoceptive attention are the Interoceptive Attention Scale (IATS) and the subscales MAIA-SR and MAIA-AR. Each of these (sub-)scale, however, measures different subcomponents of interoceptive attention. While the IATS evaluates whether individuals pay their attention “most of the time” towards bodily states indicating homeostatic disturbances (or hypervigilance towards body needs), the MAIA subscales evaluate the ability to sustain and control attention to body sensations and the ability to regulate distress through bodily attention. Based on the results, and to simplify, we could thus conclude that alexithymia is associated with more maladaptive interoceptive attention and less adaptive interoceptive attention. This conclusion further demonstrates that “interoceptive attention” covers different subcomponents and should not be uniformly assigned to multiple questionnaires (THISQ, IATS, BAQ, and MAIA subscales), especially when their items do not refer to attention.

Besides these concerns, the manuscript is excellent. Once my comments have been further addressed, I recommend its publication.

Reviewer #2: Thank you to the authors for their thorough response to my review. I believe the paper is both clearer and more concise.

I can happily recommend this paper for publication. I look forward to citing it.

A few remaining comments below:

1.Regarding the use of “neutral” – I still struggle a bit with this. I think it’s valid to relay that the authors of the THISQ used the term, but I’m not sure the term is useful in the context of your larger discussion. Specifically, it’s not clear to me that-on-the-whole, people have great access to physiological sensations that are “neutral.” Given the neurobiology of the system and available data, it seems moreso that most “typical” physiological phenomena are not experienced within conscious awareness. I’m also thinking of the newer ESM paper by Geoff Bird’s group, demonstrating that most often when people report attending to bodily sensations, they also report small increases in negative affect. All-in-all, I can’t imagine I’m the only one that will get distracted by the term “neutral.” Perhaps the distinction is more-so “self-reported perception of body sensations at rest (e.g., normal breathing) versus at times of physiology strain or exertion (e.g., dyspnea)?”

2.Lines 95-96, I would recommend using a different phrasing besides “heartbeat tracking tasks”. This phrasing is often used to describe heartbeat counting or tapping tasks. For both these tasks, I’ve seen disagreement in the literature about whether they belong in the “accuracy” or “sensibility” categories. I would maybe recommend instead citing “heartbeat detection or discrimination tasks.”

3.Lines 104-107 (and elsewhere in the manuscript)—I would place the different constructs in quotations to clarify that these are terms introduced by the meta-analyzed work, and not terms that are necessarily endorsed by the authors. This would also help to differentiate when you’re engaging in “review” verses “inference.”

4.Very nit-picky, but on 114-115, I may also recommend including “sickness” (e.g., chills or fever) or “injury” (e.g., bruising) as bodily signals represented in common self-report measures of interoceptive ability. I say this because sickness in particular, unlike hunger, thirst, fatigue, or cardiovascular sensations are not typically discussed in work on interoception (and is not objectively measured by behavioral tasks). This is one place where I think refinement of our construct is vital.

5.Line 565-566, please include a parenthetical indicating where these “weak associations” can be found (e.g., “see Table X).

Reviewer #3: The authors carefully addressed my concerns and revised their manuscript thoroughly. Well done to the authors on this solid piece of work.

7. PLOS authors have the option to publish the peer review history of their article (what does this mean?). If published, this will include your full peer review and any attached files.

Reviewer #1: No

Reviewer #2: No

Reviewer #3: **Yes: **Jessica L. Hazelton

---

## [Author Response · Author response to Decision Letter 1]

22 Aug 2024

Reviewer #1 

I would like to thank the authors for their exhaustive answers. I, nevertheless, have some remaining comments.

1. The methodology employed to assess inter-rater reliability is overall good, but also reveals some areas for potential improvement. For the title and abstract screening phase, the reported kappa values (κ = 0.10-0.39) indicate slight to fair agreement following the most liberal guidelines (Landis & Koch, 1977), but poor (Fleiss, 1981; Altman, 1991) following more conservative ones. This means that the replicability of this selection phase is not optimal and should be acknowledged in the limitations. For the extraction phase, the percentage agreement shows a high level of reliability (84.4 to 90.6%), which is commendable, but the reliance on percentage agreement alone, without considering chance agreement, may overestimate true reliability. This should also be acknowledged in the limitations.

Many thanks to the reviewer for this observation. We agree that these aspects of inter-rater reliability were suboptimal and have acknowledged this in the limitations of the revised manuscript (L 1212-1215).

2. Although I have explained that the THISQ and the BAQ are not measures of interoceptive attention, the authors stand by their interpretation based on the following arguments: (1) The authors of the THISQ (Vlemincx et al., 2023) note that THISQ scores correlated moderately with other measures of non-emotional interoceptive attention, such as the BAQ and the Noticing scale of the MAIA; (2) The BAQ is currently classified as a self-report of interoceptive attention in the literature (e.g., Trevisan et al. 2020; Vlemincx et al. 2023), as noticing and frequency of awareness involves orienting attention to salient interoceptive signals (Mehling, 2012). Here are my answers: (1) the authors of the THISQ wrote: “Self-reported interoception as assessed by the THISQ is defined as the self-reported awareness or observation of these sensations, assessed by the rate at which persons notice or feel (changes in) these internal bodily sensations. By assessing the ‘noticing’ and ‘feeling’ of sensations, we excluded interoceptive features such as discrimination, accuracy and attention, to avoid specific emotional or cognitive interpretations of sensations (similar to the BAQ and BPQ)” (p.1238). They also wrote: “… the THISQ assesses self-reported interoception... However, it does not explicitly assess interoceptive accuracy, nor interoceptive attention… The THISQ items are not subjective evaluations of whether one pays attention to these sensations, nor of whether one perceives these sensations accurately.” (p.1249) A moderate correlation does not support the conclusion that two measures evaluate the same construct. It is expected that a measure of attention correlates with a measure of awareness, as more attention should lead to more awareness; the two constructs are related but distinct. Put differently, while attention is necessary for individuals to notice and be aware of their sensations, it is not sufficient. (2) For the BAQ, it is more debatable as the original authors have themselves qualified this questionnaire as a measure of interoceptive attention. Again, however, although I agree that attention would help to better notice or predict bodily sensations, I do not agree that the BAQ is a direct measure of interoceptive attention given that no item refers to attention per se.

We greatly appreciate these comments regarding the THISQ. On balance, classification of this questionnaire is indeed challenging—particularly in the context of its overall very weak association with alexithymia. As the Reviewer rightfully notes (and we thank them for including the excerpt from the original article), Vlemincx et al. (2023) regard the THISQ as measuring “the self-reported awareness or observation of these sensations, assessed by the rate at which persons notice or feel (changes in) these internal bodily sensations.” We personally feel that classification of the THISQ as a measure of interoceptive awareness is not suitable, given the issues with this umbrella term in the literature, described in the ‘Interoceptive constructs and measurement’ section of the Discussion (p. 48). 

In their previous comments regarding the THISQ, the Reviewer proposed that “neutral interoceptive detection would be more relevant” for classification of the THISQ, which is more specific and encouraged in the manuscript. Due to the valid conceptual issues raised by the Reviewer, we made the decision to relabel ‘interoceptive accuracy’ to ‘interoceptive sensing’. This decision was in accordance with how Desmedt et al. (2023, pp. 14-15) have defined this construct: “the sense of internal signals by the nervous system across conscious and nonconscious levels…Interoceptive sensing may be underpinned by interoceptive detection (called “interoceptive accuracy” by Garfinkel et al., 2015), interoceptive magnitude (Khalsa et al., 2018), and interoceptive localization (called “interoceptive discrimination” by Khalsa et al., 2018)”. We feel that this re-classification, whilst broader in terms of the included and applicable constructs, encompasses the Reviewer’s perspectives on classification of the THISQ. 

Given the amendment of (adaptive and maladaptive) ‘interoceptive accuracy’ initially proposed to ‘interoceptive sensing’, this necessitated review, and in some instances, reclassification of other questionnaires. One such scale was the Noticing scale from the MAIA, currently regarded as an aspect of interoceptive attention in the literature. We agreed with the points made in the Reviewer’s third comment regarding underlying subfactors of interoceptive attention. Noticing MAIA seem more in line with the proposed construct of ‘interoceptive sensing’, as the items pertain to the basic detection of comfortable, neutral, and uncomfortable sensations, which may implicitly involve and be strengthened by interoceptive attention, but is ultimately not dependent on this process—nor do items explicitly refer to attentional control toward sensations.

Classification of the BAQ is particularly troublesome, given the classification of it as both a measure of (potentially maladaptive) interoceptive attention characterised by somatisation in the literature (e.g., Trevisan et al., 2021), and within the authors’ description of the scale (Shields et al., 1989). It is unfortunate that meta-analysis could not be conducted on the four subscales comprising the BAQ (i.e., Note Response/Changes in Body Process”: ability to read one's own body reaction to food, fatigue, weather, and changing in energy levels; “Predict Body Reaction”: ability to predict one's own body state from actual body signal; “Sleep-Wake Cycle”: bodily signals related to circadian rhythm; “Onset of Illness”: ability to recognise illnesses signals). Given that the BAQ has remained a point of concern for the Reviewer, we recognise that our initial reply to their comment did not adequately address this. Accordingly, qualitative item analysis was conducted to determine what subjective interoceptive aspects/beliefs are measured by BAQ items. We identified that the items covered various interoceptive domains, including detection (noticing subtle changes in bodily signals), accuracy (correctly identifying interoceptive signals), prediction (forecasting future bodily states based on current sensations and environmental factors), and differentiation (distinguishing between different bodily states). Following this review, we agree with the Reviewer—the BAQ should not be conceptualised as a self-report measure of (adaptive) interoceptive attention. We have therefore re-classified the BAQ has a measure of adaptive interoceptive sensing but note that the scale also contains measurement of predictive ability (e.g., onset of illness, predict body reaction). We feel that this these subjective beliefs are possibly in line with ‘interoceptive memory’, as conceptualised by Desmedt et al. (2023, p.14), “involving any memory process related to internal signals”. Such capacities can potentially be understood within predictive accounts of interoception, as the ability to anticipate and prepare for future body states and needs is inherently based on past experiences and current interoceptive and environmental cues. Past experiences and memory plays a key role in this process, as it enables consolidation and encoding, enabling individuals to predict internal bodily sensations and needs based on their prior experiences deemed relevant for homeostasis and allostasis (e.g., Barrett et al. 2016, https://www.affective-science.org/pubs/2016/active-inference-allostasis-ptb.pdf).

Accordingly, the manuscript has been adjusted to reflect these substantive changes, which are peppered throughout the Abstract, Introduction, and primarily, the Discussion section to reflect this revision, particularly in the following sections: Key differences in interoceptive self-reports, Implications, Conclusion.

3. This debate about the construct (i.e., whether or not it is interoceptive attention) measured by certain questionnaires is not without consequences. Indeed, the authors conclude that alexithymia is associated with more interoceptive attention. I do not agree. Questionnaires relevant to interoceptive attention are the Interoceptive Attention Scale (IATS) and the subscales MAIA-SR and MAIA-AR. Each of these (sub-)scale, however, measures different subcomponents of interoceptive attention. While the IATS evaluates whether individuals pay their attention “most of the time” towards bodily states indicating homeostatic disturbances (or hypervigilance towards body needs), the MAIA subscales evaluate the ability to sustain and control attention to body sensations and the ability to regulate distress through bodily attention. Based on the results, and to simplify, we could thus conclude that alexithymia is associated with more maladaptive interoceptive attention and less adaptive interoceptive attention. This conclusion further demonstrates that “interoceptive attention” covers different subcomponents and should not be uniformly assigned to multiple questionnaires (THISQ, IATS, BAQ, and MAIA subscales), especially when their items do not refer to attention.

We thank the Reviewer for again sharing their valuable insights and perspectives. Interoceptive attention is a particularly nebulous construct, given that various definitions and categorisations of what constitutes attention vastly differ across the literature (e.g., Khalsa et al., 2018, Desmedt et al., 2018, Murphy et al., 2019, Suksasikp & Garfinkel, 2022). Firstly, we wish to confirm that we concluded alexithymia is associated with maladaptive interoceptive attention, per the associations with IATS. Although the manuscript alluded to this scale tapping into hypervigilance, we have reiterated this in the revised discussion. Moreover, we initially concluded that lower alexithymia is associated with more adaptive interoceptive attentional beliefs. In the Discussion section, we have provided a brief clarification following discussion of the ‘adaptive’ aspects in discussion of the associations with global alexithymia to advise that the converse applies: less adaptive interoceptive beliefs are associated with higher alexithymia (L 846-848).

Subsequent to our revision of accuracy to ‘interoceptive sensing’, we note that MAIA-Noticing and BAQ are no longer classified as measures of interoceptive attention in the revised manuscript. Additionally, MAIA-BL has been re-classified as a measure of adaptive interoceptive interpretation, as items regard the use and interpretation of bodily signals to guide decision-making, making sense of or resolving emotional distress, and belief that the body serves as a valuable source of emotional information. The revised classification of the interoceptive attention construct now includes three adaptive measures (MAIA-AR, MAIA-SR, MAIA-ND), and one maladaptive measure (IATS). We retained MAIA-ND as a measure of adaptive interoceptive attention, as Desmedt et al. (2023) have previously contended that this measure is an aspect of attention, coupled with reverse-scoring of items suggesting that higher ND scores are akin to attending to and accepting sensations of pain and discomfort, relative to distraction and avoidance—aspects underpinning maladaptive interoceptive attention styles (Khalsa et al., 2018). These changes have been made to the manuscript, and peppered throughout the Discussion section.

We are inclined to concur with Desmedt et al., in that these subjective beliefs, assessable via self-report, could broadly considered as measures of interoceptive attention. However, as Desmedt and the Reviewer note, these measures should not be regarded as tapping into the same aspect of attention (or sensing, interpretation, and memory) until evidence for convergence is established. This is a notion that we completely agreed with in the original drafting and first revision. However, it is clear from the Reviewer’s comments that this conclusion was not explicitly phrased. We have adjusted the Discussion, specifically the Implications section, to express this (L 1157-1161), and again, thank the Reviewer for highlighting this important contention. 

We have provided an overview of previous and current classifications at the end of this document.

Besides these concerns, the manuscript is excellent. Once my comments have been further addressed, I recommend its publication.

Many thanks to the Reviewer for their insights during this process. We believe that we have adequately considered and addressed their concerns. The overall comments have enabled us to greatly refine the proposed construct validity framework for interoceptive self-report scales, resulting in clearer and more specific classifications.

Reviewer #2

Thank you to the authors for their thorough response to my review. I believe the paper is both clearer and more concise. I can happily recommend this paper for publication. I look forward to citing it.

Many thanks to the Reviewer! Their comments have greatly improved the clarity of the paper and further in reducing confusion regarding the THISQ.

A few remaining comments below:

1.Regarding the use of “neutral” – I still struggle a bit with this. I think it’s valid to relay that the authors of the THISQ used the term, but I’m not sure the term is useful in the context of your larger discussion. Specifically, it’s not clear to me that-on-the-whole, people have great access to physiological sensations that are “neutral.” Given the neurobiology of the system and available data, it seems moreso that most “typical” physiological phenomena are not experienced within conscious awareness. I’m also thinking of the newer ESM paper by Geoff Bird’s group, demonstrating that most often when people report attending to bodily sensations, they also report small increases in negative affect. All-in-all, I can’t imagine I’m the only one that will get distracted by the term “neutral.” Perhaps the distinction is more-so “self-reported perception of body sensations at rest (e.g., normal breathing) versus at times of physiology strain or exertion (e.g., dyspnea)?”

Many thanks to the Reviewer for alerting us to the ESM paper! We have incorporated those findings into discussion of the IATS (L 983), as this is the scale the study utilised to operationalise interoceptive attention. We have classified the IATS as a measure of maladaptive interoceptive attention, given that its items are indicative of hypervigilance and homeostatic disturbance, which may explain the observed association between increased attention in daily life with increased negative valence and fatigue. 

In addressing the comments regarding our conceptualisation of the THISQ as a measure of neutral interoceptive attention from Reviewer 1, we have re-classified the THISQ as a potential measure of adaptive interoceptive sensing. Interoceptive sensing has replaced the interoceptive accuracy construct previously proposed. This domain, in accordance with Desmedt et al.’s (2023, pp. 14-15) conceptualisation of the construct, involves: “the sense of internal signals by the nervous system across conscious and nonconscious levels…Interoceptive sensing may be unde

---

## [Editor Report · Decision Letter 2]

2 Sep 2024

A systematic review and meta-analysis of the relationship between subjective interoception and alexithymia: Implications for construct definitions and measurement

PONE-D-24-17726R2

Dear Dr. Van Bael,

We’re pleased to inform you that your manuscript has been judged scientifically suitable for publication and will be formally accepted for publication once it meets all outstanding technical requirements.

Kind regards,

Carlos Miguel Martins Campos

Academic Editor

PLOS ONE
---

## [Editor Report · Acceptance letter]

8 Sep 2024

PONE-D-24-17726R2 

PLOS ONE

Dear Dr. Van Bael, 

I'm pleased to inform you that your manuscript has been deemed suitable for publication in PLOS ONE. Congratulations! Your manuscript is now being handed over to our production team.

Kind regards, 

on behalf of

Dr. Carlos Miguel Martins Campos 

Academic Editor

PLOS ONE